# Non-autistic persons modulate their speech rhythm while talking to autistic individuals

Tatsuya Daikoku[1,2,3]*, Shinichiro Kumagaya[4], Satsuki Ayaya[4], Yukie Nagai[2,5]

1 Graduate School of Information Science and Technology, The University of Tokyo, Tokyo, Japan,
2 International Research Center for Neurointelligence (WPI-IRCN), UTIAS, The University of Tokyo, Tokyo, Japan, 3 Center for Brain, Mind and KANSEI Sciences Research, Hiroshima University, Hiroshima, Japan, 4 Research Center for Advanced Science and Technology, The University of Tokyo, Tokyo, Japan, 5 Institute for AI and Beyond, The University of Tokyo, Tokyo, Japan

* daikoku.tatsuya@mail.u-tokyo.ac.jp

**Data Availability Statement:** All data are available in the Supporting Information files and in the Open Science Framework (https://osf.io/ayn2w/).

## Abstract

How non-autistic persons modulate their speech rhythm while talking to autistic (AUT) individuals remains unclear. We investigated two types of phonological characteristics: (1) the frequency power of each prosodic, syllabic, and phonetic rhythm and (2) the dynamic interaction among these rhythms using speech between AUT and neurotypical (NT) individuals. Eight adults diagnosed with AUT (all men; age range, 24–44 years) and eight age-matched non-autistic NT adults (three women, five men; age range, 23–45 years) participated in this study. Six NT and eight AUT respondents were asked by one of the two NT questioners (both men) to share their recent experiences on 12 topics. We included 87 samples of AUT-directed speech (from an NT questioner to an AUT respondent), 72 of NT-directed speech (from an NT questioner to an NT respondent), 74 of AUT speech (from an AUT respondent to an NT questioner), and 55 of NT speech (from an NT respondent to an NT questioner). We found similarities between AUT speech and AUT-directed speech, and between NT speech and NT-directed speech. Prosody and interactions between prosodic, syllabic, and phonetic rhythms were significantly weaker in AUT-directed and AUT speech than in NT-directed and NT speech, respectively. AUT speech showed weaker dynamic processing from higher to lower phonological bands (e.g. from prosody to syllable) than NT speech. Further, we found that the weaker the frequency power of prosody in NT and AUT respondents, the weaker the frequency power of prosody in NT questioners. This suggests that NT individuals spontaneously imitate speech rhythms of the NT and AUT interlocutor. Although the speech sample of questioners came from just two NT individuals, our findings may suggest the possibility that the phonological characteristics of a speaker influence those of the interlocutor.

## Introduction

### Characteristics of speech in autism spectrum disorder

Speech is essential for human social communication. Speech rhythm plays an important role in speech intelligibility [1] and can convey the speaker's communicative intent and emotion

**Funding:** This research was supported by JST CREST 'Cognitive Feelings' (Grant Number: JPMJCR21P4), JSPS KAKENHI (Grant Number 22KK0157, 22H05210, 21H05063, 21H05053, 20K22676), JST Moonshot Goal 9(JPMJMS2296), Institute for AI and Beyond, the University of Tokyo, and World Premier International Research Centre Initiative (WPI), MEXT, Japan. The funders had no role in study design, data collection and analysis, decision to publish, or preparation of the manuscript.

**Competing interests:** The authors have declared that no competing interests exist.

[2, 3]. Speech rhythms comprise prosody, syllable, and phoneme. Inconsistency in speech rhythm between speakers causes misunderstandings and miscommunication [4]. For example, autistic (AUT) individuals (i.e., with a clinical diagnosis of autism spectrum disorder) have specific speech rhythm characteristics [5, 6]. AUT is a developmental disorder characterised by differences in social skills, communication, and repetitive patterns of behaviour [7]. A previous study claimed that a specific speech rhythm in AUT individuals impedes the smooth communication with non-autistic individuals [8]. Speech prosody, characterised by intonation, accentuation, and stress, is often weak and monotonous in AUT individuals (hereafter, 'AUT speech') [6] and has been typically utilised as a diagnostic marker of AUT [9]. Previous studies have reported important associations between prosodic performance and better communication between AUT and neurotypical (NT) individuals [10, 11]. Thus, the prosodic rhythm in AUT speech may impede smooth communication with NT individuals.

## Characteristics of speech directed to specific persons

In the last decade, studies increasingly focused on the characteristics of speech directed to a specific person. Evidence has shown that the speech of a specific person, such as AUT individuals and infants, as well as that of an NT adult directed to a specific person have specific speech rhythm characteristics.

For example, adult speech directed to infants and children, referred to as infant-directed and child-directed speech, respectively, is characterised by prosodic exaggeration, while adult speech directed to adults (adult-directed speech) has a strong syllable rhythm [12, 13]. Importantly, prosodic exaggeration contributes to speech intelligibility and competence and facilitates an infant's learning of human languages [2]. Further, infants prefer infant-directed speech with prosodic exaggeration to adult-directed speech with less prosody [14]. They also learn prosody first and speak based on the learned knowledge on speech rhythm. This suggests that adult speakers intentionally or unintentionally adjust the prosody strength in speech for the listener's preference and speech learnability [12]. By this fine-tuning of speech rhythm, infant and child listeners can better understand what adult speakers are talking about and correctly react to them [14].

This evidence raises the question of how AUT speech characteristics (i.e., weak prosody) influence NT speech directed to AUT individuals. In the past, numerous studies have revealed AUT speech characteristics [6, 10]. Despite the rich evidence on AUT speech, the characteristics of NT speech directed to AUT individuals (hereafter, 'AUT-directed speech') remain unclear. In previous studies with infant- and foreigner-directed speech [15], speech modulation (vowel hyperarticulation) has always been accompanied by heightened positive affect. Such speech modulation can occur not only in the absence of heightened positive affect in infant-directed speech, but even in the presence of heightened negative affect in adult foreigner-directed speech [15]. Thus, NT speakers adapt their speech to address the needs of the target audience. It is possible that NT individuals modulate their speech while talking to adult AUT individuals.

## Phonological hierarchy in speech rhythm

The speech rhythm forms a nested 'phonological hierarchy' of prosody, syllable, and phoneme [16, 17]. Such phonological hierarchy can be identified from the amplitude modulation (AM) hierarchy of speech waveforms below approximately 40 Hz. Evidence has shown that AM hierarchy of speech signals, including prosody, syllable, and phoneme, can be detected in the temporal structure of speech waveforms such as delta (<4 Hz), theta/alpha (4–12 Hz), and beta/low gamma (12–40 Hz), respectively, regardless of speech types [13, 18].

Linguistically, such phonological hierarchy is classically represented as a tree that captures the relative prominence of units [16, 19] (Fig 1). In the tree representation of linguistic hierarchy (Fig 1A), a 'parent' node (element) at one level of the hierarchy encompasses ≥1 'daughter' nodes at a lower hierarchy level. Adjacent connections between parent and daughter nodes are indicated as 'branches' in the tree. An example is given in Fig 1B and 1C: in the sentence '*kyou wa ii tenki desu*', a parent node, such as the word '*tenki*' at the prosodic level, would have three daughter nodes at the syllabic level, comprising the three syllables or moras (i.e., 'te', 'n', and 'ki'). The acoustic underpinning of this linguistic hierarchy is represented in an AM hierarchy (Fig 1D). For example, prosodic rhythm of the '*tenki*' is reflected in the frequency band below 4 Hz, and syllabic rhythms of the 'te', 'n', and 'ki' are reflected in the frequency band around 4–12 Hz.

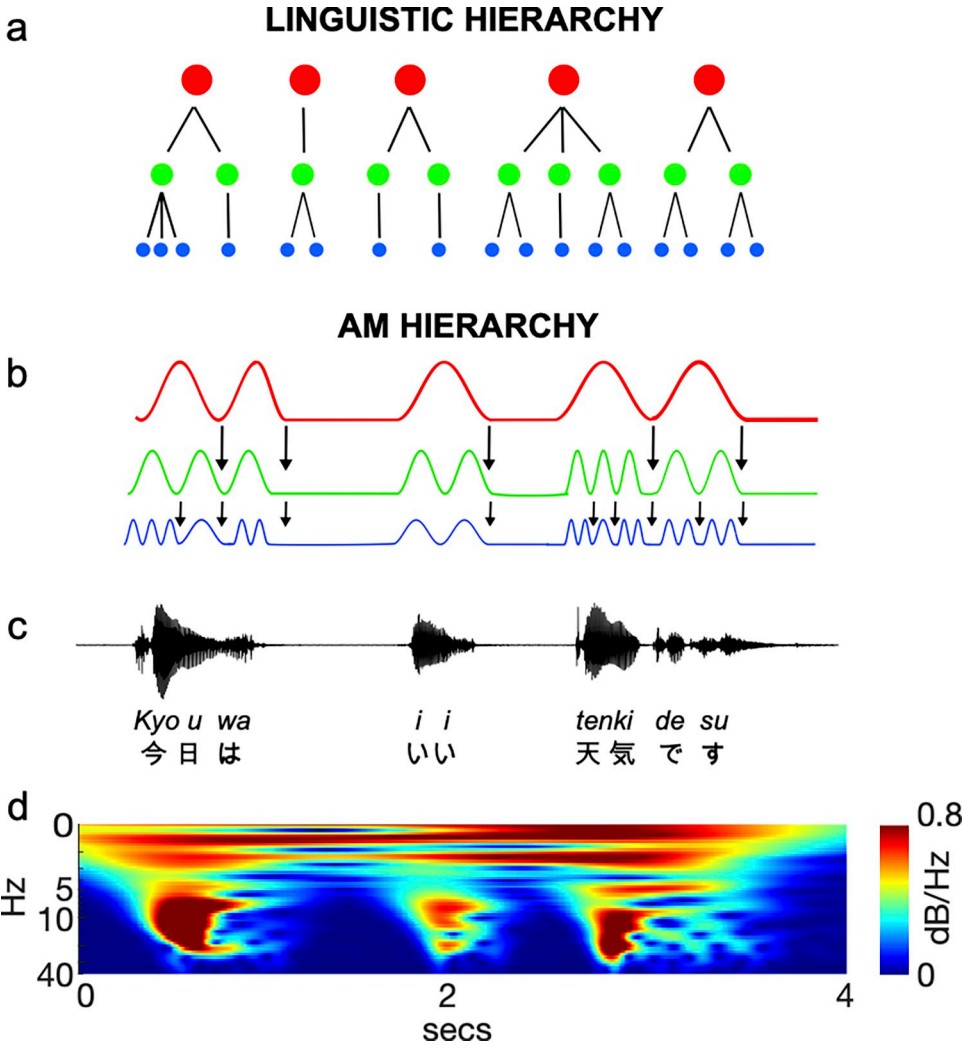

**Fig 1. Hierarchical rhythm structure in speech.** The speech rhythm could be hierarchically organised by either letters (**a**) or amplitude modulation (AM) (**b**) based on the speech waveform (**c**). Circles (**a**) and waveforms (**b**) with red, green, and blue colours represent prosody, syllable, and phoneme, respectively. Scalograms depict the AM envelopes derived by recursive application of probabilistic amplitude demodulation (**d**). A continuous wavelet transform was run on each AM envelope from randomly chosen 4-s speech excerpts from a neurotypical individual. The x-axis denotes time (4 s), and the y-axis denotes the modulation rate (0.1–40 Hz). All speech signals were normalized based on z-scores (i.e., mean = 0, SD = 1).

The prosodic, syllabic, and phonetic bands in the AM hierarchy interact with each other. For example, temporal alignment of modulation peaks between the prosodic and syllabic AM bands and influences the intelligibility of speech rhythm patterns [20]. Furthermore, the prosodic–syllabic phase synchronisation is greater for spontaneous speech by literate adults than for speech by illiterate adults [21]. Importantly, phonological AM hierarchy can be parsed from large (e.g., prosody) and smaller (e.g., syllables) linguistic units [22], suggesting dynamical processing between different speech rhythms. Thus, previous evidence implies that not only the strong (exaggerated) prosody itself but also the strong dynamical processing between different phonological hierarchy levels (e.g., between prosodic and syllabic rhythms) contributes to speech intelligibility and competence.

### Purpose of the present study

In this study, we examined how NT persons modulate their speech rhythm while talking to AUT individuals. During speech recording, six NT and eight AUT adults (respondents) were asked by one of two NT adults (questioners) to share their recent experiences on 12 topics (sweetness, gloom, etc.). To extract the AM envelopes of each prosody, syllable, and phoneme in the phonological hierarchy, we applied probabilistic amplitude demodulation (PAD) based on Bayesian inference [23]. We analysed the frequency power of the AM envelopes of prosody, syllable, and phoneme using fast Fourier transform (FFT) and the interactions and dynamics among the prosodic, syllabic, and phonetic rhythms using a transfer entropy analysis, which captures the dynamic properties between two variables [24].

First, we investigated how prosodic, syllabic, and phonetic powers in respondents' speech vary depending on their AUT diagnosis. We predicted that prosodic power would be weaker in AUT speech than in NT speech, in agreement with previous evidence [6]. Further, we examined how dynamical interaction among the prosodic, syllabic, and phonetic rhythms in AUT speech differs from that among the rhythms in NT speech. Previous evidence suggests that the interaction between different phonological rhythms (e.g., prosody vs. syllable) reflects the intelligibility of speech rhythm patterns [20, 21], implying that the interaction of prosodic (delta) and syllabic (theta) rhythms is key to understanding the speech characteristics [25]. Further, it has been shown that the weak interaction between prosodic and syllable rhythms can be detected when the power of prosodic rhythm is weak [12, 18]. From these findings, we hypothesised that AUT speech that often exhibits weak prosody also shows a weak dynamical interaction between different phonological rhythms.

Second, we investigated how the characteristics of phonological hierarchy in respondents' speech correlate with those in questioners' speech. We predicted that respondents' speech rhythm would influence the speech rhythm of the questioners and that the phonological hierarchy of the questioners' speech would become similar to that of the respondents (Fig 2). It is possible that their prosody becomes weaker in AUT-directed speech than in NT-directed speech. Furthermore, we predicted a weaker interaction among the prosodic, syllabic, and phonetic rhythms in AUT-directed speech than in NT-directed speech. If so, the results would suggest that phonological hierarchy in AUT speech influences the way of speaking to AUT individuals. NT individuals may spontaneously adapt their phonological characteristics to those of AUT speech.

## Materials and methods

### Data collection

Eight adults diagnosed with AUT (all men; age range, 24–44 years) and eight age-matched non-autistic adults (three women, five men; mean age, 23–45 years) participated in this study. They were all native Japanese speakers and recruited widely through a Japanese website. The

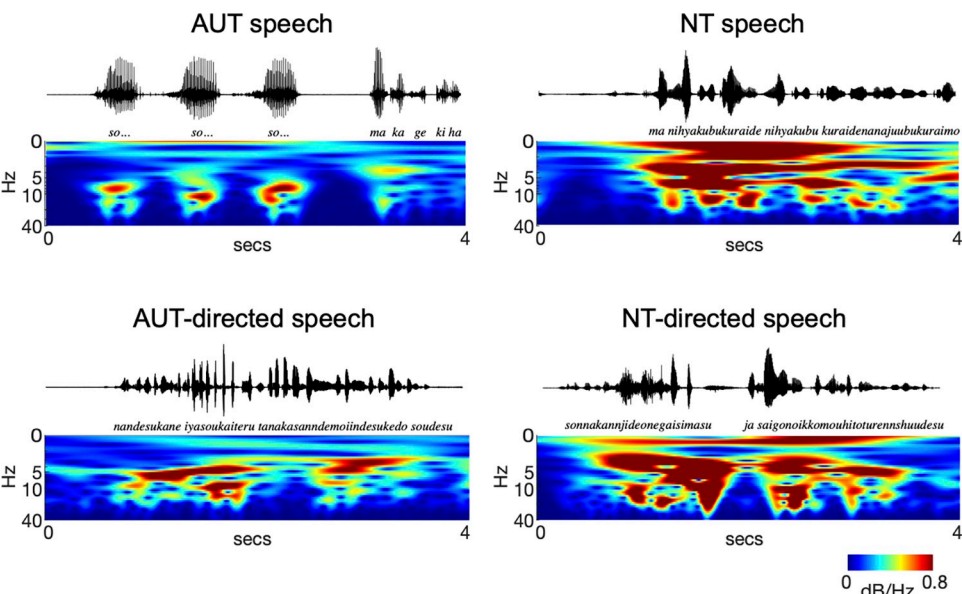

**Fig 2. Scalograms depicting the amplitude modulation envelopes derived by recursive application of probabilistic amplitude demodulation.** A continuous wavelet transform was run on each amplitude modulation (AM) envelope from randomly chosen 4-s excerpts of AUT speech, NT speech, AUT-directed speech by NT persons, and NT-directed speech by NT persons. The x-axis denotes time (4 s), and the y-axis denotes the modulation rate (0.1–40 Hz). The maximal amplitude is normalised to 0 dB. Note that the low frequency structure (<5 Hz) visible in NT and NT-directed speech is absent in AUT and AUT-directed speech. AUT, autism; NT, neurotypical.

AUT diagnosis was made by a psychiatrist according to the Diagnostic and Statistical Manual of Mental Disorders (DSM-5 [26]). AUT is not only a diagnosis, but also an individual characteristic, and therefore, the Multi-dimensional Scale for Pervasive Developmental Disorder and Attention Deficit/Hyperactivity Disorder (MSPA) was also used to assess the characteristics of developmental disabilities in general. Furthermore, the Wechsler Adult Intelligence Scale-III (WAIS-III) and Wechsler Memory Scale-revised (WMS-R) were used to measure intellectual level and memory function. The above behavioural assessments took about 3 h in total. A simpler questionnaire was also used to assess autism spectrum disorder (ASD) characteristics. Specifically, the Autism Spectrum Quotient [27], Adult Self-Report [28], and Autism Spectrum Screening Questionnaire [29] were used to measure ASD characteristics. The publicly available data about characteristics of the participants were summarized and have been deposited to an external source (https://osf.io/ayn2w/?view_only=d87230eaeaea428f94464bf633ae4118). The study was conducted in accordance with the guidelines of the Declaration of Helsinki and was approved by the Ethics Committee of the University of Tokyo. All participants were informed the content of the interview and safety of this investigation and were assured that their data would be protected. The NT questioner was aware that they were talking to an AUT individual, while they were not aware that the purpose of the study was determining how their speech changed. They provided written informed consent to participate in this study. All participants received compensation for their participation.

An NT questioner and either an NT or AUT respondent participated in a single interview (Fig 3). Questioners and respondents had no chance to communicate with each other before the interview. All eight AUT participants were respondents, while six NT participants and the other two NT participants were respondents and questioners, respectively. In each pair, both individuals knew if the partner had AUT or not. During speech recording (Marantz, PMD661MKII MP-REC-002, sampling rate = 44.1 kHz, bit rate = 16 bit) through a

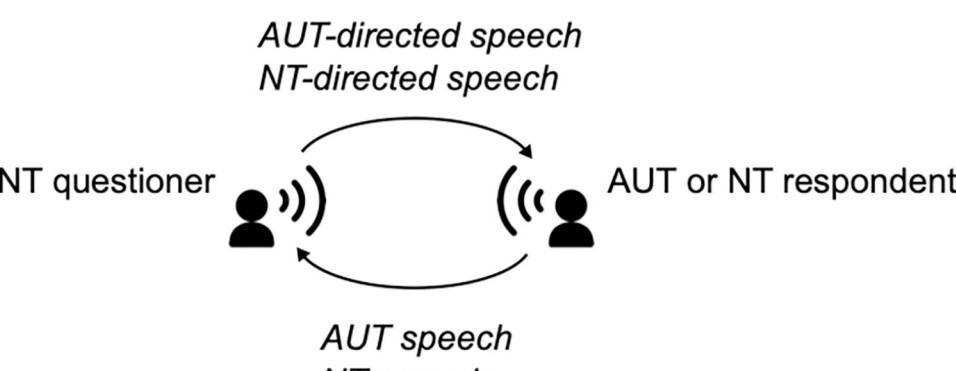

**Fig 3. Overview of the interview design of the present study.** An NT questioner and either an NT or AUT respondent participated in a single interview. The NT questioner asked the AUT or NT respondent to share their recent experiences on the following 12 topics using their episodic memory: sweetness, insults, loneliness, gloom, clients, hate, transportation, kindness, agile or prompt, ginger, attractiveness, and thoughtfulness. Then, the AUT and NT respondents reported their experience on each topic. The speech by the NT questioner directed to an AUT or NT respondent was referred to as '*AUT-directed speech*' or '*NT-directed speech*', respectively. The speech by an AUT or NT respondent directed to the NT questioner was referred to as '*AUT speech*' or '*NT speech*', respectively. AUT, autism; NT, neurotypical.

condenser-type head-worn microphone (SHURE, model BETA54), the six NT and eight AUT respondents were asked by one of two NT questioners to share their recent experiences on 12 topics: sweetness, insults, loneliness, gloom, clients, hate, transportation, kindness, agile or prompt, ginger, attractiveness, and thoughtfulness. Interviews were conducted inside a sound-proof room to block background noise. The AUT or NT respondent thus reported their experience on each topic using their episodic memory. Each interview finished within 30 min.

The length of each speech sample was >10-s with 2-s silent periods at both the beginning and end of the sample. Because samples were very short and varied in length, the variations between samples could influence the time-frequency analysis. Speech samples in which the length was <10-s were excluded from the analysis. The speech by an NT questioner directed to AUT and NT respondents was referred to as 'AUT-directed speech' and 'NT-directed speech', respectively, and that by AUT and NT respondents directed to the NT questioner was referred to as 'AUT speech' or 'NT speech', respectively. We analysed all speech sounds by the NT questioners (i.e., AUT- and NT-directed speech by NT persons) and NT and AUT respondents (AUT and NT speech). Eventually, 87 samples of AUT-directed speech, 72 of NT-directed speech, 74 of AUT speech, and 55 of NT speech were included in the analysis (Table 1).

## Data analysis

This study examined the characteristics of phonological AM hierarchy in AUT and NT speech and AUT-directed and NT-directed speech. We applied a Bayesian inference-based PAD to

**Table 1. Number of samples and mean duration of NT and AUT speech (NT and AUT respondents) and NT-directed and AUT-directed speech (NT questioners).**

|  | NT speech | AUT speech | NT-directed speech | AUT-directed speech |
|---|---|---|---|---|
| **Sample (n)** | 55 | 74 | 72 | 87 |
| **Sample duration (s)** | 16.9±1.87 | 13.36±1.35 | 20.30±1.04 | 24.01±0.83 |

Averages are reported as mean ± standard error of the mean.

NT, neurotypical; AUT, autistic spectrum

extract phonological AM hierarchy from speech sound [23]. Acoustic speech signals essentially comprise temporal information, contributing to rhythmic structures, and spectral information, contributing to pitch and other acoustic features. The PAD model extracts a cascade of AM envelopes (temporal information) at different oscillatory rates (delta, theta/alpha, and beta/gamma) from spectral information using Bayesian inference.

First, for a qualitative analysis, we analysed speech signal scalograms (Fig 2). Typically, studies have used a spectrogram to depict the spectrum of a sound changing through time, representing the quickly varying spectral feature of speech waveforms (S4 Appendix, middle). The spectrogram uses a constant-length window, which is shifted in time and frequency but does not oscillate [18]. Because the spectrogram uses a constant window, the time-frequency resolution of the spectrogram is fixed. In contrast, the scalogram is the time-frequency representation of the continuous wavelet transform (CWT) of a sound. The CWT uses a wavelet window, which is shifted in time and frequency, and oscillates. The scalogram can be more useful than the spectrogram to understand slow varying temporal feature (S4 Appendix, bottom).

The scalograms depict the temporal feature of AM envelopes derived by the recursive application of PAD. CWT was run on each AM envelope from randomly chosen 4-s excerpts taken from total speech signals (>10 second) of AUT, NT, AUT-directed, and NT-directed speech, and NT speech. Due to the scalogram being longer, it is difficult to qualitatively unveil each level of the phonological rhythm hierarchy (i.e., prosody, syllable, and prosody) and the relationship between the different levels of the phonological hierarchy. In Fig 2, the x-axis denotes time (0–4 s), and the y-axis denotes the modulation rate (0.1–40 Hz). The low frequency structure (<5 Hz) visible in NT and NT-directed speech was absent in AUT and AUT-directed speech. Thus, the scalograms support our hypothesis of similarities in the phonological characteristics between AUT and AUT-directed speech and between NT and NT-directed speech.

Next, based on the qualitative understanding using scalogram, we decided the frequency index in AM envelopes (prosody: 1–4 Hz, syllable: 4–12 Hz, phoneme: 12–40 Hz), and quantitatively analysed two types of speech characteristics using three AM envelopes. Previous research analysed both adult-directed speech and infant-directed speech [12] and detected a similar frequency index for prosody, syllable, and phoneme in the AM hierarchy. Thus, from the qualitative analyses based on scalogram in this study, we confirmed these frequency indices can be applied to the analysis of adult-directed speech.

The first was the frequency power of prosodic, syllabic, and phonetic rhythms in AUT and NT speech using FFT. The FFT is a computational tool that facilitates power spectrum analysis [30] and can analyse signals in the frequency domain. The second type of characteristic consisted of the dynamic interactions among prosodic, syllabic, and phonetic rhythms in AUT and NT speech. We applied a transfer entropy analysis to understand these interactions. The transfer entropy is a non-parametric statistic that can measure the dynamic properties of two variables [24]. The transfer entropy could answer a question on the directionality of information dynamics from a specific phonological hierarchy, such as prosody, syllable, and phoneme, to another hierarchy. The oscillators are dynamically modulated from slower to faster bands in a top-down manner [22, 30]: delta oscillators modulate the theta oscillators, and theta oscillators modulate the gamma oscillators. Such a 'cascade' oscillatory system is thought to contribute to encoding the phonological AM hierarchy and thus parsing of large (e.g., prosody) and smaller (e.g., syllables) linguistic units in a top-down manner [22]. The transfer entropy can verify such information dynamics: a high value of transfer entropy from prosodic to syllabic AM envelopes suggests that core speech rhythm is transferred from prosodic to syllabic levels.

The acoustic signals were first normalised based on the z-score (mean = 0, SD = 1) in case the sound intensity influenced the spectrotemporal modulation feature. The spectrotemporal

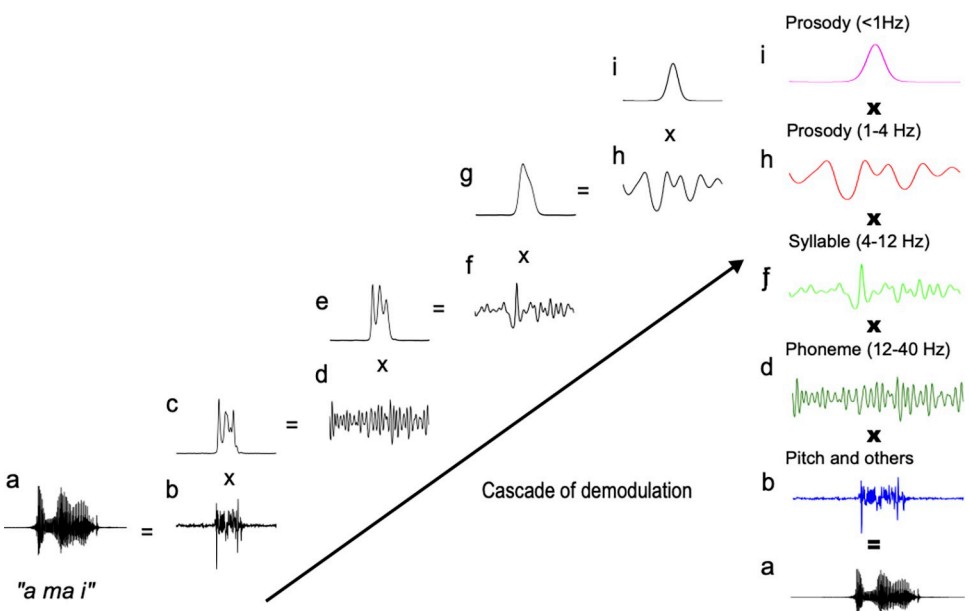

**Fig 4. Signal processing steps in the probabilistic amplitude demodulation (PAD) model.** Example of an amplitude modulation (AM) hierarchy derived by recursive PAD application. In the first demodulation round (left column), the data (**a**) are demodulated using PAD set to a fast time scale. This yields a relatively rapidly varying envelope (**b**) and a carrier (**c**). In the second demodulation round (middle column), the demodulation process is re-applied using a slower time scale than before. This yields a slower (**d**) and a faster (**e**) daughter envelope. Daughter envelopes (**d**) and (**e**) form the two levels of the resulting AM hierarchy (right column). Mathematically, these two levels (**d** and **e**) can be multiplied back by the first carrier (**c**, bottom left) to yield the original signal (**a**).

modulation of the signals was analysed using PAD [23] to derive the dominant AM patterns. In acoustic engineering, the speech signals can be expressed as the product of the slow varying AM patterns and the rapidly varying carrier or frequency modulation (FM) patterns [5, 23]. AM patterns are implicated in fluctuations in sound intensity, which underlie a primary acoustic correlate of perceived rhythm based on onset timing, prosody, syllable, and phoneme. In contrast, FM patterns reflect fluctuations in spectral frequency and noise [23]. It is possible to separate the AM envelopes of speech signals from the FM structure by amplitude demodulation processes. The PAD model infers the modulators and a carrier based on Bayesian inference and identifies the envelope that best matches the data and *a priori* assumptions. PAD can be run recursively using different demodulation parameters each time to generate a cascade of amplitude modulators at different oscillatory rates: low delta (<1 Hz), delta (1–4 Hz), theta/alpha (4–12 Hz), and beta/low gamma (12–40 Hz) bands, forming an AM hierarchy isolated from a carrier at a higher frequency rate (>40 Hz) (Fig 4). More specifically, amplitude demodulation is the process by which a signal ($y_t$) is decomposed into a slowly varying modulator ($m_t$) and a rapidly varying carrier ($c_t$):

$$y_t = m_t * c_t \tag{1}$$

PAD [23] implements amplitude demodulation as a process of learning and inference. Learning corresponds to the estimation of parameters that describe distributional constraints, such as the expected variation timescale of the modulator. Inference corresponds to the estimation of the modulator and carrier from the signals based on learned or manually defined parametric distributional constraints. This information is encoded probabilistically in the

likelihood: $P(y_{1:T}|c_{1:T}, m_{1:T}, \theta)$, prior distribution over the carrier: $p(c_{1:T}|\theta)$, and prior distribution over the modulators: $p(m_{1:T}|\theta)$. Here, the notation $x_{1:T}$ represents all the samples of the signal x, running from 1 to a maximum value T. Each of these distributions depends on a set of parameters $\theta$, which controls factors such as the typical timescale of variation of the modulator or the frequency content of the carrier. For more detail, the parametrised joint probability of the signal, carrier, and modulator is:

$$P(y_{1:T}, \ c_{1:T}, \ m_{1:T}|\theta) = P(y_{1:T}|c_{1:T}; m_{1:T}, \ \theta)*p(c_{1:T}|\theta)*p(m_{1:T}|\theta) \quad (2)$$

Bayes' theorem is applied for inference, forming the posterior distribution over the modulators and carriers, given the signal:

$$P(c_{1:T}, \ m_{1:T}|y_{1:T}, \ \theta) = P(y_{1:T}, \ c_{1:T}, \ m_{1:T}|\theta)/P(y_{1:T}|\theta) \quad (3)$$

The full solution to PAD is a distribution over the possible pairs of modulators and carriers. The most probable pair of modulator and carrier given the signal is returned:

$$m*_{1:T}, \ c*_{1:T} = argmax \, P(c_{1:T}, \ m_{1:T}|y_{1:T}, \ \theta) \quad (4)$$

That is, PAD estimates the most appropriate modulator (envelope) and carrier, based on Bayesian inference, that identifies the envelope that best matches the data and *a priori* assumptions. The solution takes the form of a probability distribution that describes the likelihood of a particular setting of modulator and carrier given the observed signal. Thus, PAD summarises the posterior distribution by returning the specific envelope and carrier with the highest posterior probability and, therefore, represents the best match to the data. PAD can be run recursively using different demodulation parameters each time, generating a cascade of amplitude modulators at different oscillatory rates to form an AM hierarchy (Fig 4). The positive slow envelope is modelled by applying an exponential nonlinear function to a stationary Gaussian process. This produces a positive-valued envelope of constant mean over time. The degree of correlation between points in the envelope can be constrained by the timescale parameters of variation of the modulator (envelope), which may either be entered manually or learned from the data. In the present study, we manually entered the PAD parameters to produce the modulators at each of the four oscillatory band levels (i.e., low delta: <1 Hz delta: 1–4 Hz, theta/alpha: 4–12 Hz, beta/gamma: 12–40 Hz) isolated from a carrier at a higher frequency rate (>40 Hz). The carrier reflects components, including noise and pitches, for which the frequencies are much higher than those of the core modulation bands in the phrase, prosodic, and syllabic phonological components. A modulation band <1 Hz (Fig 4I) was excluded from the analysis because the frequency band reflecting phonological features, including prosody, syllable, and phoneme, ranges from ~1 Hz to ~40 Hz [13], and temporal waveform <1 Hz generally reflects sentence-level rhythm (duration >1 s). In each speech sample, the modulators (envelopes) of the three oscillatory bands (delta: 1–4 Hz, theta/alpha: 4–12 Hz, beta/gamma: 12–40 Hz) were converted into frequency domains by FFT. The spectral modulator analysis reflects the speed of sound intensity fluctuation over time. High modulation frequency corresponds to fast modulations and vice versa (Fig 4). The modulation spectra were averaged across all samples in each group (AUT, NT, AUT-directed, and NT-directed speech).

We also examined the dynamic interactions among the prosodic, syllabic, and phonetic rhythms by transfer entropy analyses. Transfer entropy is a non-parametric statistic that measures dynamical properties between two variables [24]. Transfer entropy from a variable X to another variable Y is the amount of uncertainty reduced in future values of Y by knowing the

past values of X given the past values of Y. The transfer entropy can also be expressed as:

$$T_{X \to Y} = H(Y_t | Y_{t-1:t-L}) - H(Y_t | Y_{t-1:t-L}, X_{t-1:t-L}) \tag{5}$$

where H(X) is the entropy of X based on information theory [31]. This analysis could answer a question on the directionality of transfer information between the level of phonological hierarchy X (i.e., prosody, syllable, and phoneme) and another level Y. Previous evidence suggests that the oscillators are hierarchically and dynamically modulated frm slower to faster bands in a top-down manner [22, 30]; the delta oscillators modulate the theta oscillators, and theta oscillators modulate the gamma oscillators. Such a 'cascade' oscillatory system is thought to contribute to encoding the phonological AM hierarchy and thus the parsing of large (e.g., prosody) and smaller (e.g., syllables) linguistic units in a top-down manner [22]. Considering the evidence, we hypothesised that the dynamical system between adjacent speech rhythm levels may play a key role in speech intelligibility. We particularly predicted that AUT and AUT-directed speech may have specific characteristics in terms of speech rhythm dynamics.

## Statistical analysis

Statistical analyses were conducted using Jamovi version 1.2 [32]. We used separate ANOVA models to compare between AUT and NT speech (Analysis 1) and between AUT-directed and NT-directed speech (Analysis 2). The ANOVA models for FFT included the within-participant factor of rhythm (i.e., prosody, syllable, and phoneme). In contrast, the ANOVA models for transfer entropy included the within-participant factor of a pair of AM levels (i.e., prosody–syllable, prosody–phoneme, and syllable–phoneme) and the within-participant factor of directionality (i.e., top–down and bottom–up).

Further, we averaged the frequency power of prosody and transfer entropy for each participant. We then performed a Shapiro–Wilk test for normality separately for the datasets containing frequency power of prosody and transfer entropy. Then, to understand whether each NT questioner changes their way of speaking depending on the speech characteristics of respondents, we performed a one-tailed Pearson's or Spearman's correlation test for each frequency power and transfer entropy between NT questioners and all of NT and AUT respondents, based on an alternative hypothesis of positive correlation. We selected $p$-values of $<0.05$ as the threshold for statistical significance and used a false discovery rate method for *post-hoc* analysis and multiple testing of significant effects.

## Results

### AUT speech and NT speech

Eight adults diagnosed with AUT (all men; age range, 24–44 years) and eight age-matched non-autistic adults (three women, five men; age range, 23–45 years) participated in this study. Six NT and eight AUT respondents were asked by one of two NT questioners (both men) to share their recent experiences on 12 topics. We included 87 samples of AUT-directed speech (from an NT questioner to an AUT respondent), 72 samples of NT-directed speech (from an NT questioner to an NT respondent), 74 samples of AUT speech (from an AUT respondent to an NT questioner), and 55 samples of NT speech (from an NT respondent to an NT questioner).

First, we examined how the characteristics of phonological AM hierarchy in AUT speech differ from those in NT speech. Using three AM envelopes corresponding to prosodic, syllabic, and phonetic rhythms, we analysed two types of speech characteristics: the frequency power of each prosodic, syllabic, and phonetic rhythm using FFT and the dynamic interactions among the prosodic, syllabic, and phonetic rhythms using a transfer entropy analysis.

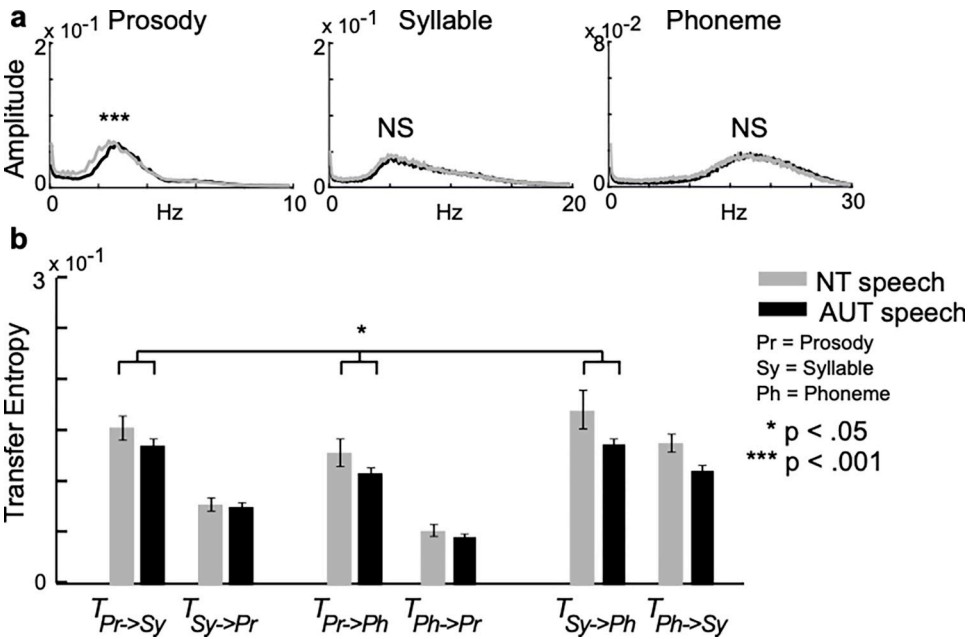

**Fig 5. Speech by individuals with autism (AUT) and neurotypical (NT) subjects.** Fast Fourier transform (**a**) showing a weaker prosodic rhythm in AUT speech than in NT speech, but no clear difference in syllabic or phonetic rhythms between AUT and NT. Transfer entropy analysis (**b**) revealed weaker top-down dynamics (e.g., prosody to syllable) of speech rhythms in the phonological hierarchy of AUT speech than of NT speech. Error bars indicate the standard deviation of the mean.

The analysis of variance (ANOVA) indicated a significant interaction between rhythm and group ($F_{[1.20, 187.83]}$ = 7.08, $p$ = 0.006, $\eta^2_p$ = 0.043; S1 Appendix). The frequency power of prosodic rhythm was significantly weaker in AUT speech than in NT speech ($p < 0.001$; Fig 5A, left). In contrast, there was no significant difference between AUT and NT speech in terms of syllabic ($p$ = 0.214; Fig 5A, middle) or phonetic rhythms ($p$ = 0.918; Fig 5A, right). The ANOVA for transfer entropy indicated a significant interaction between directionality and group ($F_{[1, 157]}$ = 3.89, $p$ = 0.049, $\eta^2_p$ = 0.024). The dynamics of speech rhythms in the phonological hierarchy from large (e.g., prosody) to smaller (e.g., syllables) linguistic units were weaker in AUT speech than in NT speech ($p$ = 0.028; Fig 5B). This suggests weaker top-down dynamics in AUT than in NT. In contrast, there was no significant difference between AUT and NT speech in terms of bottom-up dynamics ($p$ = 0.283).

### AUT-directed and NT-directed speech

To understand how NT persons modulate their speech rhythm while talking to AUT individuals, we also examined how the characteristics of phonological AM hierarchy in NT speech directed to AUT (i.e., AUT-directed speech) differ from those to NT (i.e., NT-directed speech) and how the characteristics in AUT-directed and NT-directed speech correlate with those in AUT and NT speech. As in the analyses of AUT and NT speech, we applied both FFT and transfer entropy analysis and found a marked similarity between AUT and AUT-directed speech and between NT and NT-directed speech. More specifically, FFT showed a weaker prosodic rhythm in AUT-directed speech than in NT-directed speech (Fig 6A, left; S2 Appendix for complete results) but no clear difference in syllabic and phonetic rhythms (Fig 6A, middle and right). Further, transfer entropy analysis showed more consistent weak dynamical

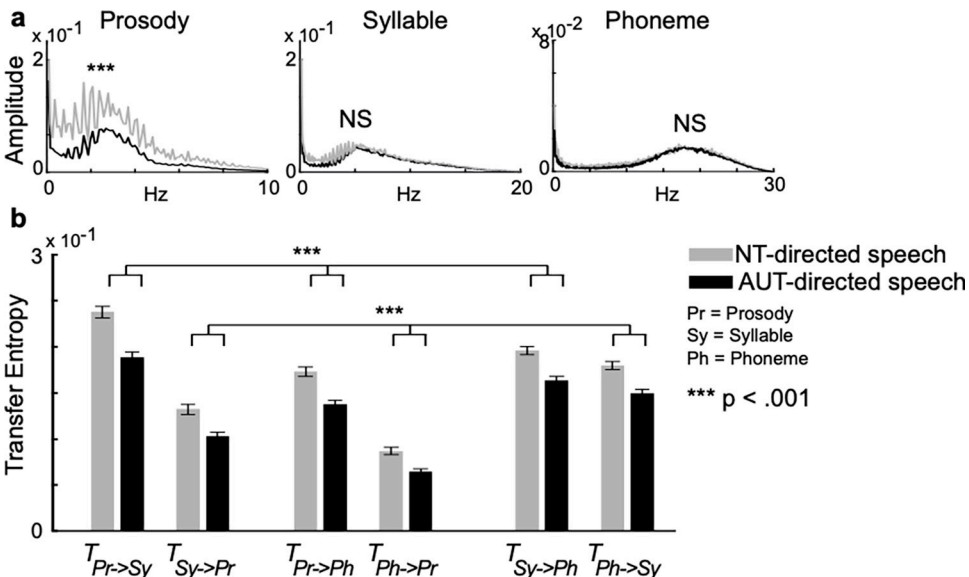

**Fig 6. Speech directed to adults with autism (AUT-directed speech) and neurotypical (NT) adults (NT-directed speech).** Fast Fourier transform (**a**) showing a weaker prosodic rhythm in AUT speech than in NT speech, but no clear difference in syllabic and phonetic rhythms. Transfer entropy analysis (**b**) showed more consistent weaker mutual dependence between different speech rhythms in AUT-directed speech than in NT-directed speech. Importantly, there is a marked similarity between AUT and AUT-directed speech and between NT and NT-directed speech (compare Fig 5). Error bars indicate the standard deviation of the mean.

interaction among the prosodic, syllabic, and phonetic rhythms in AUT-directed speech than in NT-directed speech regardless of directionality (Fig 6B).

These observations were reflected by ANOVA, which indicated a significant interaction between rhythm and group ($F_{[1.20, 187.83]}$ = 7.08, $p$ = 0.006, $\eta^2_{\,p}$ = 0.043; see S2 Appendix for detailed results). The frequency power of prosodic rhythm was significantly weaker in AUT-directed speech than in NT-directed speech ($p < 0.001$; Fig 6A, left). In contrast, there was no significant difference between AUT-directed and NT-directed speech in syllabic ($p$ = 0.214; Fig 6A, middle) or phonetic rhythms ($p$ = 0.918; Fig 6A, right). The ANOVA for transfer entropy indicated a significant interaction between rhythm and group ($F_{[1.57, 199.95]}$ = 3.71, $p$ = 0.036, $\eta^2_{\,p}$ = 0.028). AUT-directed speech had significantly weaker mutual dependence between prosody and syllable ($p < 0.001$), prosody and phoneme ($p < 0.001$), and syllable and phoneme ($p < 0.001$) components than NT-directed speech.

### Correlation test

The ANOVA results (Figs 5 and 6) detected a similarity between AUT and AUT-directed speech and between NT and NT-directed speech in terms of frequency power of prosody and dynamical interaction (transfer entropy) among the prosodic, syllabic, and phonetic rhythms. This implies that a NT questioner may utilize a similar way of speaking to the NT and AUT respondents. However, due to the small sample size there is little diversity in AUT and NT, and it may be difficult to show exactly how NT subjects adapt to AUT utterances. Therefore, to understand whether each individual NT questioner changes their way of speaking depending on the speech characteristics of respondents, we further performed Spearman's correlation tests before Shapiro–Wilk normality test for frequency power of prosody and transfer entropy among the prosodic, syllabic, and phonetic rhythms. The data of frequency powers and transfer entropy were averaged for each respondent (8 AUT and 6 NT) and corresponding

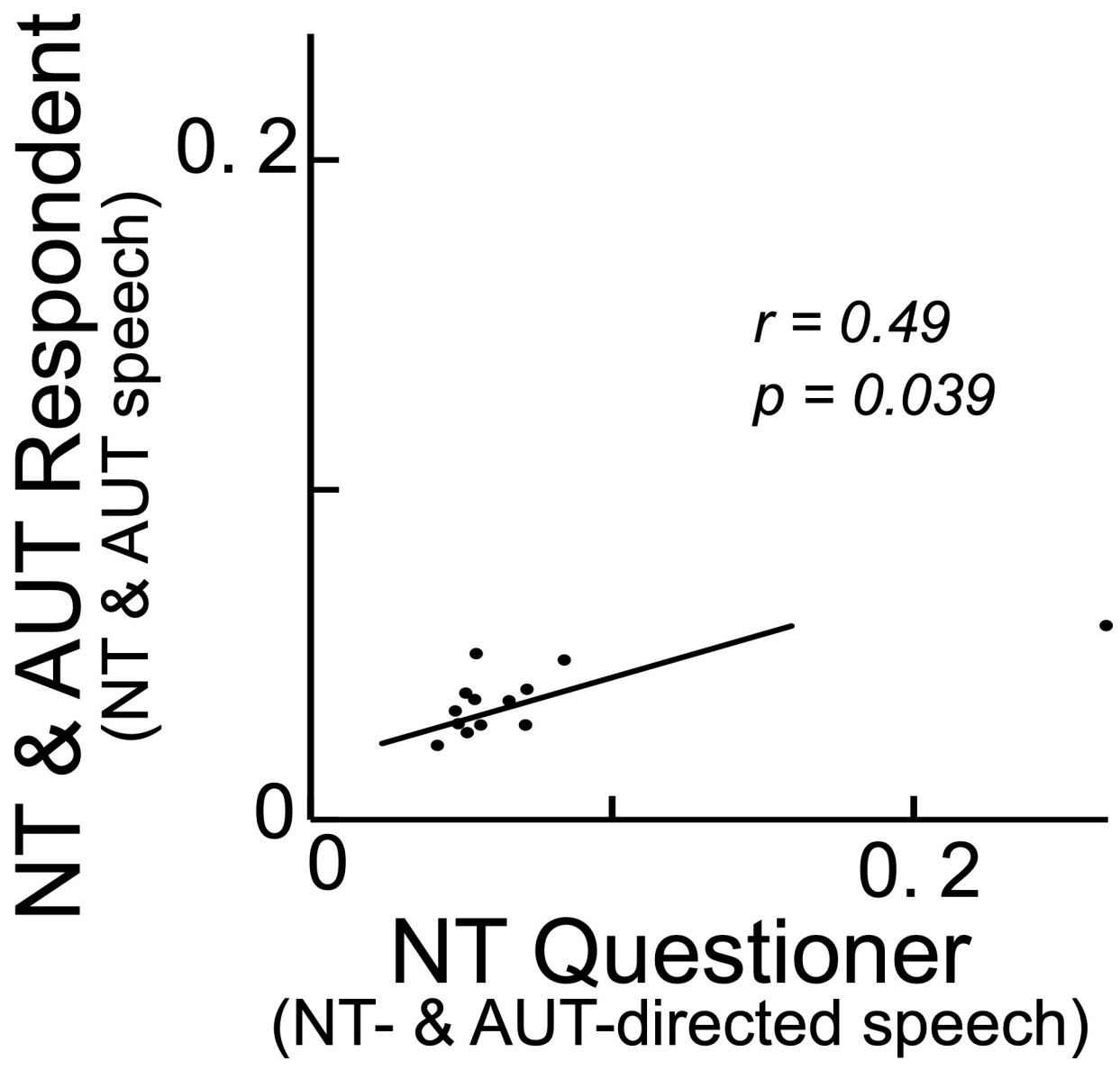

**Fig 7. Correlation of frequency power of prosodic rhythm between a NT questioner and NT and AUT respondent.**

questioner (2 NT). The results showed significant positive correlation of the frequency power of prosody ($rs = 0.49$, $p = 0.039$) (Fig 7). No other significant correlation for frequency power of prosody and transfer entropy were detected (S3 Appendix). These findings suggested that each NT questioner may change their way of speaking depending on the prosodic characteristics of responders.

## Discussion

### Characteristics of prosody are shared between AUT speech and AUT-directed speech

The present study aimed to reveal how NT persons modulate their speech rhythm while talking to AUT individuals. We investigated two types of speech rhythm characteristics: (1) the

frequency power of each prosodic, syllabic, and phonetic rhythms and (2) the dynamic interaction among these rhythms, using speech between AUT and NT individuals. We found similarities in phonological characteristics between AUT and AUT-directed speech and between NT and NT-directed speech. The prosody was significantly weaker in AUT-directed speech than in NT-directed speech and in AUT speech than in NT speech, whereas no significant group difference was found in syllabic and phonetic rhythms. Further, the weaker the frequency power of prosody in NT and AUT respondents (i.e., NT and AUT speech), the weaker the frequency power of prosody in NT questioners (i.e., NT- and AUT-directed speech).

In general, speech prosody is adjusted based on the type and age of the listener. For example, infants show a preference for infant-directed speech with prosodic exaggeration over adult-directed speech with reduced prosody [14]. They also learn a prosodic property for speech first and speak based on the learned prosody. Accordingly, adults typically exaggerate prosody (i.e., motherese) when speaking to an infant or child [12]. However, such a speech characteristic is modulated by a type of listener's auditory skill. For example, the deficiency of prosodic perceptual skills, which is often observed in AUT individuals, modulates not only their own speech properties (i.e., weak prosody [6]), but also the speech properties of the conversation partner [33–35]. A previous study examined how children with AUT react to child-directed speech compared to NT children [36], showing that children with AUT paid less attention to child-directed speech with prosodic exaggeration than did NT children. The difference in preference, attention, and auditory skill in the listener may cause the speaker to change the way of speaking. Notably, our study showed that compared to syllable and phoneme, the prosody characteristics were similar between AUT speech and AUT-directed speech. Prosody may be particularly important for emotional and smooth communication [37, 38]. A speaker may facilitate smooth communication with a listener by imitating speech rhythm characteristics that reflect listener individuality.

## Motivation of adjusting the way of speaking: Learnability and imitation

A previous study has suggested that NT speakers adapt their foreigner-directed as well as infant-directed speech to address the needs of the target audience [15]. It is known that an NT speaker's speech characteristics are modulated based on the listener's speech characteristics and preference. NT speakers modulate the prosody strength in speech for the infant listener's preference and speech intelligibility for enhancing speech learnability [14]. However, for AUT adult listeners, NT speakers may modulate their speech for smooth communication rather than learnability, which is a different motivation from speech modulation of child- or infant-directed speech.

One hypothesis is that the weak prosody of AUT-directed speech may come from the imitation or mimicry of AUT speech with weak prosody. Evidence has shown that imitation occurs during communication [39]. For example, people spontaneously imitate other people's facial gestures [40] and facial expressions [41]. Moreover, during conversation, a person tends to mirror another person's posture [42–44] and to imitate the other's speech patterns [45, 46], vowel sounds [47], vocal intensity [48], and speaking rate [49], and to move in synchrony with the other's speech rhythms [50–52]. Such imitation, mirroring (or motor mimicry), and synchronisation behaviours are important for enhancing affinity between persons [53], feeling empathy [54], and facilitating communication [55].

The prosody exaggeration of child-/infant-directed speech has been suggested both to explain natural selection for human language from an anthropological perspective [56] and to facilitate the learning of the phonological structure of human language in infants [57]. The exaggeration of prosody facilitates learning [14]. Hence, adults typically exaggerate prosody

(i.e., motherese) when speaking to an infant or child [12] to support the listener's learning. Meanwhile, the present study found weak prosody when speaking to individuals whose speech shows weak prosody, which may have a detrimental effect on learning. However, imitation may contribute to synchronisation and better communication [58, 59]. Our findings suggest that speakers may imitate the listener's prosody for smooth communication.

Another hypothesis is that the AUT respondents might imitate the way of speaking of the questioner. Although previous research has shown that AUT individuals exhibit deficits in imitation [59, 60], there is little evidence for dysfunction of the mirroring system in AUT. In a review by Hamilton [61], social 'top-down' modulation of mirroring systems has been suggested to be a critical factor in AUT. In our study, each NT questioner talked with both AUT and NT respondents, while AUT respondents talked with only one NT questioner. That is, AUT respondents did not talk with other AUT individuals. Consequently, whether their speech pattern is already adapted when talking to AUT individuals remains unclear. Thus, we could not compare between NT-directed and AUT-directed speech by AUT individuals, and the present imitation findings apply to NT individuals only. Further studies are warranted to investigate how AUT speech is modulated when speaking to AUT individuals compared to NT individuals.

To our knowledge, this study is first to unveil the phonological hierarchy in AUT by speech signal scalograms. Fig 2 clearly shows that the low frequency structure (<5 Hz) visible in NT and NT-directed speech is absent in AUT and AUT-directed speech. Thus, the evidence from the scalogram also supports our hypothesis of similarities in phonological characteristics between AUT and AUT-directed speech and between NT and NT-directed speech.

## Dynamic interaction among different hierarchies in speech rhythms

The present study further revealed weaker dynamic interactions among the prosodic, syllabic, and phonetic rhythms in AUT speech than in NT speech, and in AUT-directed speech than in NT-directed speech. These interactions are associated with the processing of 'branches' in the tree of linguistic hierarchy (Fig 1A). In this respect, a previous study showed that the temporal alignment of modulation peaks between different phonological bands influences speech intelligibility [20]. That is, a strong or stressed syllable is perceived when delta and theta modulation peaks are in alignment. The placement of stressed syllables governs metrical patterning in speech. Particularly, the prosodic–syllabic (i.e., delta–theta) phase synchronisation in spontaneous speech has been reported to be greater in literate adults than in illiterate adults [21], implying that the phase alignment of prosodic (delta) and syllabic (theta) rhythms is key to understanding the language-learning brain [25]. For example, individual differences in sensitivity to delta- and theta-AM rates (i.e., prosodic and syllabic rhythms) are associated with language development disorders [25, 62]. We found a weaker interaction between speech rhythms in AUT speech than in NT speech, and in AUT-directed speech than in NT-directed speech. This may indicate that individual differences in interaction between speech rhythms also occur in AUT because of abnormal prosodic characteristics, such as weakness and monotony [6].

In addition, we found weaker speech rhythm dynamics from higher to lower phonological bands (e.g., from prosody to syllable) in AUT speech than in NT speech. The neurophysiological interpretation of AM patterns of speech signals by EEG is relatively straightforward. In the auditory cortex, different band neural oscillators phase-synchronise with different AM patterns in speech signals at matching rates [63–65]. This oscillatory synchronisation contributes to parsing speech signals into their respective phonological units [66]. For example, delta, theta, and beta/gamma EEG oscillators in the auditory cortex contribute to the perception of

prosodic, syllabic, and phonetic information, respectively [65, 67, 68]. Importantly, these oscillators are hierarchically modulated from slower to faster bands [22, 26]; that is, delta oscillators modulate theta oscillators, and theta oscillators modulate gamma oscillators. Such cascaded oscillatory modulation is thought to contribute to the encoding of phonological AM hierarchy and parsing of larger (e.g., prosody) and smaller (e.g., syllables) linguistic units in a top-down manner [22]. This evidence suggests that individual differences in the top-down interaction of speech rhythms are implicated in language ability. Our findings indicate that the weak top-down interaction of speech rhythms in the phonological AM hierarchy may also be associated with AUT. This study, however, did not control for sex differences between groups. Thus, it is possible that sex-specific speech characteristics influenced the findings [69].

## Limitations and future research

The main limitation of this study is that characteristics of speech signals can be influenced by multiple factors other than phonology: frequency, word length, grammatical complexity, and insertions of pauses and fillers. Furthermore, possible confounders could include social anxiety or depression (common comorbidities in AUT individuals), affinity to the topic being discussed (AUT individuals often have problems with 'small talk'), whether AUT were camouflaging or masking during the interaction, what background knowledge/expectations the NT individuals had about AUT, and whether participants were nervous (i.e., if participants had participated in studies before). Another limitation is that although this study used a sufficient sample for the statistical analysis methods (Table 1), and all samples were sufficiently gathered from all speakers in terms of numbers and durations, this study used a small sample size of individuals. Particularly, each respondent was asked by only one of the two NT questioners. Further, this study could not control for sex (gender) differences between groups. This suggests that as well as AUT-specific characteristics, participant-specific characteristics may have influenced our findings. Nevertheless, given the small sample size in this study, we also conducted correlation analysis to examine whether each individual NT questioner changes their way of speaking depending on the speech characteristics of respondents. The findings indicated the prosodic powers in NT questioners were positively correlated with those in NT and AUT respondents. Hence, this study suggested the possibility that an NT individual may change their way of speaking depending on the prosodic characteristics of interlocutor.

This study also checked the MSPA to control the characteristics of developmental disabilities in general. Furthermore, the WAIS-III and WMS-R were used to control intellectual level and memory function. From these participants, this study detected weak prosody in AUT as a general and typically known characteristic of AUT speech. Further, we detected novel findings of weak dynamical interaction between prosodic and syllabic rhythms using analysis of transfer entropy. This finding may help elucidate latent components that characterise various types of speech.

## Conclusion

The present study suggested the core characteristics of the phonological hierarchy in AUT and AUT-directed speech and showed similarities in phonological characteristics between the two. The frequency power in the prosodic AM band was weaker in AUT speech and AUT-directed speech than in NT speech and NT-directed speech. We also found that the weaker the frequency power of prosody in NT and AUT respondents (i.e., NT and AUT speech), the weaker the frequency power of prosody in NT questioners (i.e., NT- and AUT-directed speech).

Further, the dynamic interactions among the prosodic, syllabic, and phonetic rhythms in AUT speech and AUT-directed speech were weaker than those in NT and NT-directed speech.

Although speech samples in the questionnaires (NT- and AUT-directed speech) come from just two NT individuals, this study suggests that the phonological characteristics of a speaker influence those of the interlocutor. Our findings may imply the possibility that NT individuals spontaneously imitate speech rhythms in AUT. Our findings support the importance of pedagogical intervention in NT individuals to facilitate smooth speech communication between NT and AUT individuals.

## Supporting information

**S1 Appendix. Results of statistical comparisons between AUT and NT speech.**
(DOC)

**S2 Appendix. Results of statistical comparisons between AUT-directed and NT-directed speech.**
(DOC)

**S3 Appendix. Normality test.**
(DOCX)

**S4 Appendix. Comparison of spectrogram and scalogram for a speech signal.**
(DOC)

## Author Contributions

**Data curation:** Shinichiro Kumagaya.

**Formal analysis:** Tatsuya Daikoku.

**Funding acquisition:** Tatsuya Daikoku, Yukie Nagai.

**Investigation:** Tatsuya Daikoku.

**Methodology:** Yukie Nagai.

**Supervision:** Shinichiro Kumagaya, Yukie Nagai.

**Validation:** Shinichiro Kumagaya.

**Writing – original draft:** Tatsuya Daikoku.

**Writing – review & editing:** Tatsuya Daikoku, Shinichiro Kumagaya, Satsuki Ayaya, Yukie Nagai.

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
