## [Decision Letter · Decision Letter 0]

2 Oct 2022

PONE-D-22-22806Non-autistic persons modulate their speech rhythm while talking to autistic individualsPLOS ONE

Dear Dr. Daikoku,

Thank you for submitting your manuscript to PLOS ONE. After careful consideration, we feel that it has merit but does not fully meet PLOS ONE’s publication criteria as it currently stands. Therefore, we invite you to submit a revised version of the manuscript that addresses the points raised during the review process.

We look forward to receiving your revised manuscript.

Kind regards,

Mitsuru Kikuchi, MD, PhD

Academic Editor

PLOS ONE

Journal Requirements:

2. Please change "female” or "male" to "woman” or "man" as appropriate, when used as a noun (see for instance https://apastyle.apa.org/style-grammar-guidelines/bias-free-language/gender).

4. Thank you for stating the following in the Funding Section of your manuscript: 

"This research was supported by JST CREST ‘Cognitive Mirroring’ (Grant Number: JPMJCR16E2) including AIP challenge program, JST CREST 'Cognitive Feelings' (Grant Number: JPMJCR21P4), Institute for AI and Beyond, The University of Tokyo, World Premier International Research Centre Initiative (WPI), MEXT, and JSPS KAKENHI Grant Number 21H05063, 20K22676, 22K17986,      22H05210, and 21H05053 Japan. The funding sources had no role in the decision to publish or prepare the manuscript."

"No"

Additional Editor Comments:

This research is very interesting, and the idea is unique. However, due to some problems such as the number of samples, this study cannot be accepted at this time. The sample size should be stated in the abstract so as not to mislead the reader.

Reviewers' comments:

Reviewer's Responses to Questions

**Comments to the Author**

1. Is the manuscript technically sound, and do the data support the conclusions?

Reviewer #1: Partly

Reviewer #2: No

2. Has the statistical analysis been performed appropriately and rigorously? 

Reviewer #1: Yes

Reviewer #2: Yes

3. Have the authors made all data underlying the findings in their manuscript fully available?

Reviewer #1: Yes

Reviewer #2: Yes

4. Is the manuscript presented in an intelligible fashion and written in standard English?

Reviewer #1: Yes

Reviewer #2: Yes

5. Review Comments to the Author

Reviewer #1: The author investigated if NT individuals modulate their speech while talking to adult AUT individuals through analysis of frequency power and dynamic interaction.

This study is novel in that it focused on conversation scenes between ASD and non-ASD people and performed utterance analysis, and is useful for examining way of communication with non-autistic people.

However, I found the description of some very important points were inadequate or completely missing. Therefore, I recommend that a major revision is warranted. I explain my concerns in more detail below. I ask that the authors specifically address each of my comments in their response.

Despite the authors noted that frequency variations are greater for AUT, only eight individuals with ASD participated in this study. Due to the small sample size of participants with ASD, there is little diversity in AUT utterances, and it would be difficult to show exactly how NT subjects adapt to AUT utterances.

According to the previous research cited by the authors, prosody, syllable, and phoneme indicators in the AM hierarchy were analyzed using samples of child directed speech in English. Therefore, it is unclear whether this frequency index can be applied to the analysis of speech to share recent experiences between adults (with ASD) in Japanese.

Minor

It would be helpful for reader to understand this research if the number of participants, gender and age of the participants in this study are written in the abstract.

Introduction

The authors mentioned that ‘AUT is a developmental disorder characterized by differences in social skills and communication’, but the definition of Autism Spectrum Disorder also includes Restricted, repetitive patterns of behavior.

Introduction

Two papers are cited on ‘associations between prosodic performance and better communication between AUT and neurotypical (NT) individuals’, although the authors describe it as ‘a paper’.

Introduction

The basis for hypothesizing weaker interactions between different phonological hierarchy levels in AUT speech than in NT speech is unclear.

Fig.2. It would be better to add Japanese syllables that match the sound waveform.

Reviewer #2: This study examined whether and how neurotypical (NT) persons would modulate their speech while talking to persons with autism spectrum syndrome (AUT). Interviews were held between NT and AUT persons and their speech was analyzed. Probabilistic amplitude demodulation (PAD) was used to extract amplitude envelopes of prosodic, syllable, and phoneme features in speech, fast Fourier transform was applied to analyze the frequency power of the envelopes, and transfer entropy analysis was performed to reveal interactions among prosodic, syllabic, and phonetic rhythms. The analyses showed similarities in the phonological characteristics of speech from and directed to AUT persons, suggesting that, as the main finding of the manuscript, NT persons seem to adjust their speech directed to AUT persons to improve communication.

The article presents a fair hypothesis based on previous, related research and a solid description of the sound analyses. If I understood correctly, however, it all boils down to differences in the speech of two (2) NT speakers, when directing speech to NT or AUT individuals. We are thus essentially dealing with the results of a case study that does not allow generalized statements, such as the last sentence of the abstract: the key speech samples all come from just 2 “others” or “NT individuals” who directed speech to AUT individuals. An analysis as to whether changes in their prosodic rhythm occurred over time, e.g., speech directed at AUT-individuals at the beginning compared to that at the end of the interviews, is also missing. This could have lent some support for speculation about imitation of the AUT individuals’ speech to improve smooth communication (e.g., P9), and this would have made the article somewhat more convincing.

Related to this, the authors presented actual demographics on the questioners and respondents only at the end of the manuscript. This is possibly due to the text format, but on P6, L2 they state “all the questioners”, suggesting a fair number of NT questioners directing speech to NT and AUT individuals. However, it took until P13 before they mention the exact number of questioners: two (2). Not doing this earlier and just focusing on sample numbers is misleading the reader, given the generalized statements made overall about speech of NT individuals directed to AUT persons. Furthermore, the AUT group (all males) and the NT group (5 males, 3 females) are composed of persons with inherently different voice characteristics (e.g. gender differences in the use of prosody), and it is unknown whether all samples were gathered equally from all speakers, so there is ample possibility that results were biased. Furthermore, there are various descriptions of the duration of the samples (see points 18 and 19 below), raising concerns about sample selection and the overall replicability of the analyses. The authors should clarify these issues or prove otherwise, and until then in my opinion the manuscript is unsuitable for publication.

Others:

1. Abstract, unnecessary spaces (L1, L4, L9) and period (L4, before “NT”).

2. Acronym AUT is used liberally meaning “autistic” (adjective) or “autistic persons” (adj + noun) as in the last sentence of the abstract (“speech rhythms in AUT”). Please fix this in a consistent way (e.g, “speech rhythms produced by AUT persons”, or “rhythms in AUT speech”, or something like that).

3. Introduction, Characteristics of speech in ASD. The explanation in between lines 6-12 is not very smooth. L9 “not simple but peculiar in various ways” is not clear; frequency variations (e.g., pitch escalation / melodic variability) are related to prosody. L6 and L12 seem similar.

4. P4, Phonological hierarchy in speech rhythm, first paragraph. Including a description in terms of delta, theta/alpha, beta/gamma rhythms would be informative, not just at the end of the paper in data analysis.

5. P4, Phonological hierarchy in speech rhythm, second paragraph. Fix the font.

6. P4, Phonological hierarchy in speech rhythm, second paragraph, L7: the fact that “tenki” has 3 syllables is confusing to readers unfamiliar with Japanese, please explain.

7. Fig 1. Caption, L3, “Scalograms depicting ...” > “depict”

8. Fig 1. Caption, P5, L2, “..speech excerpts by a neurotypical individuals” > “..excerpts from NT individuals”.

9. Fig 1. Caption, P5, L3. “The maximal amplitude is normalized to 0 dB”. For the sake of replicability, please explain how.

10. P3, Caption Fig 3 and else. What do the error bars represent?

11. P10, Dynamic interaction, L2, “This suggests that AUT speech and AUT-directed speech share not only......and phonetic rhythms.” Because there are weaker dynamic interactions among the rhythms in AUT speech and AUT-directed speech, they share the same rhythm and have the same interactions? This is too strong a statement, given that there is no analysis to support this.

12. P11L10, “This study did not control...between groups” > “sex (gender) differences between groups”. Note that until here, still the reader does not know the respondents’ / questioners’ gender. Given the high predominance of male AUT individuals, one would expect that all were male.

13. P11 Limitations, first sentence, “..speech signals are influenced..., could all affect the speech signals. Correct the grammar.

14. P11 Limitations, L8, “Nevertheless, this study used a sufficient sample for the statistical methods”. However, the main problem is that the key samples come from the same two questioners who directed speech to AUT persons.

15. P12, Data Collection. Here is the first time we know that the NT groups consists of 3 females and 5 males, while the AUT group consists of 8 males. What is the gender of the questioners, who provide the key speech samples? In Limitations (P11) one would at least expect a discussion as to whether results could be interpreted in terms of gender differences in the use of prosody or voice range between males and females.

16. P12, Data Collection, L3 from bottom, P13L3, P13L4, Caption Fig 5, and possibly elsewhere: “experimental paradigm” / “experiment”. Liberal use of “experiment”. An experiment requires dependent and independent variable(s) and allows inference of causal statements. There was no “experiment” here, there were interviews / talks between people.

17. P13L13, “alacrity”. There is probably a more common synonym for this.

18. P13, last paragraph, L1: “The length of each speech sample was > 10 s. For the sake of replicability of the analysis, the caption of Fig 2 states that there were 4-s excerpts. Were these randomly extracted from the samples of > 10 s? Why, because in this case you could have used samples of at least 4 s, instead of 10 s? Please provide explanation.

19. P14, Data analysis, last paragraph: “The scalograms...chosen 30-s excerpts of speech. In Fig 2, the x-axis denotes time (30 s)...Now it really gets confusing. Both Fig 2 and S3 (which actually states “S1 Appendix”) show scalograms for 4 seconds. Please provide clear and consistent descriptions.

20. Other than the sampling frequency, there is no description of recording equipment, background noise, etc.

21. P15, 2nd paragraph, “...in case the loudness influenced the spectrotemporal modulation feature”. Four lines below this: “in loudness or sound intensity”. Two lines below this: “in pitch and noise”. Pitch and loudness are psychological constructs, frequency and intensity refer to physics, which is likely meant here.

6. PLOS authors have the option to publish the peer review history of their article (what does this mean?). If published, this will include your full peer review and any attached files.

Reviewer #1: No

Reviewer #2: No

---

## [Author Response · Author response to Decision Letter 0]

24 Jan 2023

RESPONSE TO EDITORS: 

We are extremely grateful for the Editors’ insightful comments on our manuscript. We have studied the comments carefully and have made the necessary corrections in our paper. Our responses to the comments are as follows.

Comment 1:

Response: Thank you for the comment. We have ensured that the manuscript meets PLOS ONE's style requirements. We have also asked a language editing service to check the writing and ensure that it meets PLOS ONE's style.

 

Comment 2:

Please change "female” or "male" to "woman” or "man" as appropriate, when used as a noun (see for instance https://apastyle.apa.org/style-grammar-guidelines/bias-free-language/gender).

Response: Thank you for the comment. We have changed the word accordingly.

Comment 3:

We note that the grant information you provided in the ‘Funding Information’ and ‘Financial Disclosure’ sections do not match. When you resubmit, please ensure that you provide the correct grant numbers for the awards you received for your study in the ‘Funding Information’ section.

We note that you have provided funding information that is not currently declared in your Funding Statement. However, funding information should not appear in the Acknowledgments section or other areas of your manuscript. We will only publish funding information present in the Funding Statement section of the online submission form. Please remove any funding-related text from the manuscript and let us know how you would like to update your Funding Statement. 

Response: Thank you very much for the comment. We have updated and confirmed them through the online system. We have also removed any funding-related text from the manuscript. All the authors declare no competing financial interests. moreover, we have amended statements in the cover letter as follows:

‘This research was supported by JST CREST 'Cognitive Feelings' (Grant Number: JPMJCR21P4), JSPS KAKENHI (Grant Number 21H05063, 20K22676, 22H05210, 21H05053), JST Moonshot Goal 9(JPMJMS2296), Institute for AI and Beyond, the University of Tokyo, and World Premier International Research Centre Initiative (WPI), MEXT, Japan. The funders had no role in study design, data collection and analysis, decision to publish, or preparation of the manuscript.’

Comment 4:

We note that you have indicated that data from this study are available upon request. PLOS only allows data to be available upon request if there are legal or ethical restrictions on sharing data publicly. For more information on unacceptable data access restrictions, please see http://journals.plos.org/plosone/s/data-availability#loc-unacceptable-data-access-restrictions. 

Response: We thank the reviewer for the comments. All the minimal anonymized data set are available in supporting information (S1, S2, and S3 Appendix). We have stated that the data set can be found under the section of Data Availability.

Comment 5:

Additional Editor Comments:

This research is very interesting, and the idea is unique. However, due to some problems such as the number of samples, this study cannot be accepted at this time. The sample size should be stated in the abstract so as not to mislead the reader.

Response: We thank the reviewer for the important comments. We stated the sample size in the abstract section so as not to mislead potential readers as follows. 

‘Eight adults diagnosed with AUT (all men; age range, 24–44 years) and eight age-matched non-autistic NT adults (three women, five men; age range, 23–45 years) participated in this study. Six NT and eight AUT respondents were asked by one of the two NT questioners (all men) to share their recent experiences on 12 topics. We included 87 samples of AUT-directed speech (from an NT questioner to an AUT respondent), 72 of NT-directed speech (from an NT questioner to an NT respondent), 74 of AUT speech (from an AUT respondent to an NT questioner), and 55 of NT speech (from an NT respondent to an NT questioner).’

 

RESPONSE TO REVIEWER 1: 

I wish to express our strong appreciation to the Reviewer for the insightful comments on our manuscript. We have studied the Reviewer’s comments very carefully and have made necessary corrections. We feel the comments have helped us significantly improve the manuscript.

Comment 1:

Despite the authors noted that frequency variations are greater for AUT, only eight individuals with ASD participated in this study. Due to the small sample size of participants with ASD, there is little diversity in AUT utterances, and it would be difficult to show exactly how NT subjects adapt to AUT utterances.

Response: We thank the reviewer for the pertinent comments. We agree with the limitation of sample size. We clearly stated the sample size in the abstract section as to not mislead potential readers. Further, we added a correlation analysis to check not only general differences between AUT-directed speech and NT-directed speech, but also whether each NT individual changes their way of speaking depending on the speech characteristics of the interlocutor. The results suggested that the prosodic powers in NT questioners were positively correlated with those in NT and AUT respondents. This implies that an NT individual may change their way of speaking depending on the prosodic characteristics of the interlocutor. We included it in the Methods section, and as a limitation in the Discussion section as follows:

Abstract section

“Eight adults diagnosed with AUT (all men; age range, 24–44 years) and eight age-matched non-autistic NT adults (three women, five men; age range, 23–45 years) participated in this study. Six NT and eight AUT respondents were asked by one of the two NT questioners (all men) to share their recent experiences on 12 topics. We included 87 samples of AUT-directed speech (from an NT questioner to an AUT respondent), 72 of NT-directed speech (from an NT questioner to an NT respondent), 74 of AUT speech (from an AUT respondent to an NT questioner), and 55 of NT speech (from an NT respondent to an NT questioner).”

Methods section

‘Further, we averaged the frequency power of prosody and transfer entropy for each participant. We then performed a Shapiro–Wilk test for normality separately for the datasets containing frequency power of prosody and transfer entropy. Then, to understand whether each NT questioner changes their way of speaking depending on the speech characteristics of respondents, we performed a one-tailed Pearson’s or Spearman’s correlation test for each frequency power and transfer entropy between NT questioners and all of NT and AUT respondents, based on an alternative hypothesis of positive correlation. We selected p-values of <0.05 as the threshold for statistical significance and used a false discovery rate method for post-hoc analysis and multiple testing of significant effects.’

Discussion section

‘Another limitation is that although this study used a sufficient sample for the statistical analysis methods (Table 1), and all samples were sufficiently gathered from all speakers in terms of numbers and durations, this study used a small sample size of individuals. Particularly, each respondent was asked by only one of the two NT questioners. Further, this study could not control for sex (gender) differences between groups. This suggests that as well as AUT-specific characteristics, participant-specific characteristics may have influenced our findings. Nevertheless, given the small sample size in this study, we also conducted correlation analysis to examine whether each individual NT questioner changes their way of speaking depending on the speech characteristics of respondents. The findings indicated the prosodic powers in NT questioners were positively correlated with those in NT and AUT respondents. Hence, this study suggested the possibility that an NT individual may change their way of speaking depending on the prosodic characteristics of interlocutor.’

Comment 2:

According to the previous research cited by the authors, prosody, syllable, and phoneme indicators in the AM hierarchy were analyzed using samples of child directed speech in English. Therefore, it is unclear whether this frequency index can be applied to the analysis of speech to share recent experiences between adults (with ASD) in Japanese.

Response: Thank you for the important comment. The previous research cited in this paper analysed both adult-directed speech and infant-directed speech (Leong et al., 2017). They then detected a similar frequency index for prosody, syllable, and phoneme indicators in the AM hierarchy. In addition, the model that we used in this study differs from that used by Leong et al. (2017). She used a model called S-AMPH and this study used a PAD model. In a previous study, we have already confirmed that the frequency index between S-AMPH and PAD is consistent in both adults and children (Daikoku and Goswami, 2022). Further, in this study we first checked the boundary as a qualitative analysis using scalogram. Then, based on the qualitative understanding, we decided the frequency index (prosody: 1‒4 Hz, syllable: 4‒12 Hz, phoneme: 12‒40 Hz). Therefore, with evidence from previous studies (Leong et al., 2017; Daikoku and Goswami, 2022) and the confirmation based on scalogram in this study, this frequency index can be applied to the analysis of adult-directed speech. We have described this in the Introduction and Methods sections as follows:

Introduction section

‘Evidence has shown that AM hierarchy of speech signals, including prosody, syllable, and phoneme, can be detected in the temporal structure of speech waveforms such as delta (<4 Hz), theta/alpha (4–12 Hz), and beta/low gamma (12–40 Hz), respectively, regardless of speech types [13,18].’

Methods section

‘Next, based on the qualitative understanding using scalogram, we decided the frequency index in AM envelopes (prosody: 1‒4 Hz, syllable: 4‒12 Hz, phoneme: 12‒40 Hz), and quantitatively analysed two types of speech characteristics using three AM envelopes. Previous research analysed both adult-directed speech and infant-directed speech [12] and detected a similar frequency index for prosody, syllable, and phoneme in the AM hierarchy. Thus, from the qualitative analyses based on scalogram in this study, we confirmed these frequency indices can be applied to the analysis of adult-directed speech.’

Leong, V., Kalashnikova, M., Burnham, D., & Goswami, U. (2017). The temporal modulation structure of infant-directed speech. Open Mind, 1(2), 78-90.

Daikoku, T., & Goswami, U. (2022) Hierarchical amplitude modulation structures and rhythm patterns: Comparing Western musical genres, song, and nature sounds to Babytalk. PloS one. 3033;17(10), e0275631

Comment 3:

It would be helpful for reader to understand this research if the number of participants, gender and age of the participants in this study are written in the abstract.

Response: We thank the reviewer for the comments. We stated them in the Abstract section as follows:

‘Eight adults diagnosed with AUT (all men; age range, 24–44 years) and eight age-matched non-autistic NT adults (three women, five men; age range, 23–45 years) participated in this study. Six NT and eight AUT respondents were asked by one of the two NT questioners (all men) to share their recent experiences on 12 topics. We included 87 samples of AUT-directed speech (from an NT questioner to an AUT respondent), 72 of NT-directed speech (from an NT questioner to an NT respondent), 74 of AUT speech (from an AUT respondent to an NT questioner), and 55 of NT speech (from an NT respondent to an NT questioner).’

Comment 4:

Introduction

The authors mentioned that ‘AUT is a developmental disorder characterized by differences in social skills and communication’, but the definition of Autism Spectrum Disorder also includes Restricted, repetitive patterns of behavior.

Response: Thank you for the comment. As indicated, we have revised this sentence in the Introduction section as follows.

‘AUT is a developmental disorder characterised by differences in social skills, communication, and repetitive patterns of behaviour’

Comment 5:

Introduction

Two papers are cited on ‘associations between prosodic performance and better communication between AUT and neurotypical (NT) individuals’, although the authors describe it as ‘a paper’.

Response: Thank you very much for letting me know the typo. We modified it in the Introduction section as follows.

‘Previous studies have reported important associations between prosodic performance and better communication between AUT and neurotypical (NT) individuals [10,11].’

Comment 6:

Introduction

The basis for hypothesizing weaker interactions between different phonological hierarchy levels in AUT speech than in NT speech is unclear.

Response: We thank the reviewer for the pertinent comments. As indicated, we specified it more clearly in the Introduction section as follows:

‘Previous evidence suggests that the interaction between different phonological rhythms (e.g., prosody vs. syllable) reflects the intelligibility of speech rhythm patterns [20,21], implying that the interaction of prosodic (delta) and syllabic (theta) rhythms is key to understanding the speech characteristics [25]. Further, it has been shown that the weak interaction between prosodic and syllable rhythms can be detected when the power of prosodic rhythm is weak [12,18]. From these findings, we hypothesised that AUT speech that often exhibits weak prosody also shows a weak dynamical interaction between different phonological rhythms.’

Comment 7:

Fig.2. It would be better to add Japanese syllables that match the sound waveform.

Response: Thank you for the comment. Although it is difficult to insert all the Japanese syllables corresponding to these waveforms on the graphs, we have done so in figure 2 as shown.

 

RESPONSE TO REVIEWER 2:

We are extremely grateful for the Reviewers’ insightful comments on our manuscript. We have studied the Reviewers’ comments carefully and have made the necessary corrections in our paper. We believe that the comments have helped us significantly improve and refine the manuscript.

Comment 1:

The authors presented actual demographics on the questioners and respondents only at the end of the manuscript. This is possibly due to the text format, but on P6, L2 they state ‘all the questioners’, suggesting a fair number of NT questioners directing speech to NT and AUT individuals. However, it took until P13 before they mention the exact number of questioners: two (2). Not doing this earlier and just focusing on sample numbers is misleading the reader, given the generalized statements made overall about speech of NT individuals directed to AUT persons. Furthermore, the AUT group (all males) and the NT group (5 males, 3 females) are composed of persons with inherently different voice characteristics (e.g. gender differences in the use of prosody), and it is unknown whether all samples were gathered equally from all speakers, so there is ample possibility that results were biased. Furthermore, there are various descriptions of the duration of the samples (see points 18 and 19 below), raising concerns about sample selection and the overall replicability of the analyses. The authors should clarify these issues or prove otherwise, and until then in my opinion the manuscript is unsuitable for publication.

Response: We thank the reviewer for the pertinent comments. As suggested, we have stated the sample size in the abstract section so as not to mislead potential readers. Further, we have specified information regarding gender in the first paragraph of the Results section so that potential readers are informed before reading the Results and Discussion sections. Then, we have clearly stated the limitation of this study in the Discussion section as mentioned below. Further, given the concern about sample size, we also conducted a correlation analysis to not only check for general difference between AUT-directed speech and NT-directed speech, but also to check whether each NT individual changes their way of speaking depending on the speech characteristics of interlocutor. The results suggested that the prosodic powers in NT questioners were positively correlated with those in NT and AUT respondents. This implies that an NT individual may change their way of speaking depending on the prosodic characteristics of interlocutor. We have also included this as a limitation in the Discussion section as follows. Finally, we have responded to all comments raised by the reviewer. Our changes are reflected in the text as follows:

Abstract section

‘Eight adults diagnosed with AUT (all men; age range, 24–44 years) and eight age-matched non-autistic NT adults (three women, five men; age range, 23–45 years) participated in this study.’

Result section

‘Eight adults diagnosed with AUT (all men; age range, 24–44 years) and eight age-matched non-autistic adults (three women, five men; age range, 23–45 years) participated in this study. Six NT and eight AUT respondents were asked by one of two NT questioners (all men) to share their recent experiences on 12 topics. We included 87 samples of AUT-directed speech (from an NT questioner to an AUT respondent), 72 samples of NT-directed speech (from an NT questioner to an NT respondent), 74 samples of AUT speech (from an AUT respondent to an NT questioner), and 55 samples of NT speech (from an NT respondent to an NT questioner).’

Discussion section

‘Another limitation is that although this study used a sufficient sample for the statistical analysis methods (Table 1), and all samples were sufficiently gathered from all speakers in terms of numbers and durations, this study used a small sample size of individuals. Particularly, each respondent was asked by only one of the two NT questioners. Further, this study could not control for sex (gender) differences between groups. This suggests that as well as AUT-specific characteristics, participant-specific characteristics may have influenced our findings. Nevertheless, given the small sample size in this study, we also conducted correlation analysis to examine whether each individual NT questioner changes their way of speaking depending on the speech characteristics of respondents. The findings indicated the prosodic powers in NT questioners were positively correlated with those in NT and AUT respondents. Hence, this study suggested the possibility that an NT individual may change their way of speaking depending on the prosodic characteristics of interlocutor.’ 

Comment 2:

Abstract, unnecessary spaces (L1, L4, L9) and period (L4, before “NT”).

Response: Thank you for the comment. We modified the unnecessary spaces throughout the manuscript.

Comment 3:

Acronym AUT is used liberally meaning “autistic” (adjective) or “autistic persons” (adj + noun) as in the last sentence of the abstract (“speech rhythms in AUT”). Please fix this in a consistent way (e.g, “speech rhythms produced by AUT persons”, or “rhythms in AUT speech”, or something like that).

Response: Thank you very much for the important comment. As indicated, we have revised it in the Abstract section as follows.

‘This suggests that NT individuals spontaneously imitate speech rhythms of the NT and AUT interlocutor.’

Comment 4:

Introduction, Characteristics of speech in ASD. The explanation in between lines 6-12 is not very smooth. L9 “not simple but peculiar in various ways” is not clear; frequency variations (e.g., pitch escalation / melodic variability) are related to prosody. L6 and L12 seem similar.

Response: We thank the reviewer for the pertinent comments. As indicated, we have revised the sentence in the section of ‘Characteristics of speech in autism spectrum disorder’ as follows.

‘Speech prosody, characterised by intonation, accentuation, and stress, is often weak and monotonous in AUT individuals (hereafter, ‘AUT speech’) [6] and has been typically utilised as a diagnostic marker of AUT [9].’

Comment 5:

P4, Phonological hierarchy in speech rhythm, first paragraph. Including a description in terms of delta, theta/alpha, beta/gamma rhythms would be informative, not just at the end of the paper in data analysis.

Response: Thank you for the comment. As suggested, we have described the information also in terms of delta, theta/alpha, beta/gamma rhythms in the section of ‘Phonological hierarchy in speech rhythm’ as follows.

‘Evidence has shown that AM hierarchy of speech signals, including prosody, syllable, and phoneme, can be detected in the temporal structure of speech waveforms such as delta (<4 Hz), theta/alpha (4–12 Hz), and beta/low gamma (12–40 Hz), respectively, regardless of speech types [13,18].’ 

Comment 6:

P4, Phonological hierarchy in speech rhythm, second paragraph. Fix the font.

Response: Thank you for the comment. We fixed the font.

Comment 7:

P4, Phonological hierarchy in speech rhythm, second paragraph, L7: the fact that “tenki” has 3 syllables is confusing to readers unfamiliar with Japanese, please explain.

Response: Thank you for the comment. As suggested, we explained it in the Introduction section as follows.

‘a parent node, such as the word ‘tenki’ at the prosodic level, would have three daughter nodes at the syllabic level, comprising the three syllables or moras (i.e., ‘te’, ‘n’, and ‘ki’).’

Comment 8:

Fig 1. Caption, L3, “Scalograms depicting ...” > “depict”

Response: Thank you for the indication. As indicated, we have revised it as follows.

‘Scalograms depict the AM envelopes derived by recursive application of probabilistic amplitude demodulation’

Comment 9:

Fig 1. Caption, P5, L2, “..speech excerpts by a neurotypical individuals” > “..excerpts from NT individuals”.

Response: We have revised it as follows.

‘speech excerpts from a neurotypical individual’

Comment 10:

Fig 1. Caption, P5, L3. “The maximal amplitude is normalized to 0 dB”. For the sake of replicability, please explain how.

Response: We thank the reviewer for the pertinent comments. We have explained it in greater detail in the caption of Figure 1 as follows.

‘All speech signals were normalized based on z-scores (i.e., mean = 0, SD = 1)’

Comment 11:

P3, Caption Fig 3 and else. What do the error bars represent?

Response: Thank you very much for the comment. We stated about the error bars in the captions of Fig 3 and Fig 4 as follows.

‘Error bars indicate the standard deviation of the mean.’

Comment 12:

P10, Dynamic interaction, L2, “This suggests that AUT speech and AUT-directed speech share not only......and phonetic rhythms.” Because there are weaker dynamic interactions among the rhythms in AUT speech and AUT-directed speech, they share the same rhythm and have the same interactions? This is too strong a statement, given that there is no analysis to support this.

Response: We thank the reviewer for the important comments. This sentence was too strong a statement. We have excluded it from the Discussion section.

Comment 13:

P11L10, “This study did not control...between groups” > “sex (gender) differences between groups”. Note that until here, still the reader does not know the respondents’ / questioners’ gender. Given the high predominance of male AUT individuals, one would expect that all were male.

Response: We thank the reviewer for the pertinent comment. As indicated, we have revised them in the Discussion section as follows. Further, we have specified the information of gender in the first paragraph of the Results section so that potential readers are aware of these factors before reading the Results and the Discussion sections as follows:

Results

‘Eight adults diagnosed with AUT (all men; age range, 24–44 years) and eight age-matched non-autistic adults (three women, five men; age range, 23–45 years) participated in this study. Six NT and eight AUT respondents were asked by one of two NT questioners (all men) to share their recent experiences on 12 topics. We included 87 samples of AUT-directed speech (from an NT questioner to an AUT respondent), 72 samples of NT-directed speech (from an NT questioner to an NT respondent), 74 samples of AUT speech (from an AUT respondent to an NT questioner), and 55 samples of NT speech (from an NT respondent to an NT questioner).’

Discussion

‘This study, however, did not control for sex differences between groups.’

Comment 14:

P11 Limitations, first sentence, “..speech signals are influenced..., could all affect the speech signals. Correct the grammar.

Response: Thank you for the comment. We modified the sentence in the Discussion section as follows.

‘The main limitation of this study is that characteristics of speech signals can be influenced by multiple factors other than phonology: frequency, word length, grammatical complexity, and insertions of pauses and fillers.’

Comment 15:

P11 Limitations, L8, “Nevertheless, this study used a sufficient sample for the statistical methods”. However, the main problem is that the key samples come from the same two questioners who directed speech to AUT persons.

Response: Thank you very much for the important comment. We more clearly stated this important problem in the Limitation section as follows.

‘Another limitation is that although this study used a sufficient sample for the statistical analysis methods (Table 1), and all samples were sufficiently gathered from all speakers in terms of numbers and durations, this study used a small sample size of individuals. Particularly, each respondent was asked by only one of the two NT questioners.’

Comment 16:

P12, Data Collection. Here is the first time we know that the NT groups consists of 3 females and 5 males, while the AUT group consists of 8 males. What is the gender of the questioners, who provide the key speech samples? In Limitations (P11) one would at least expect a discussion as to whether results could be interpreted in terms of gender differences in the use of prosody or voice range between males and females.

Response: We thank the reviewer for the pertinent comments. The sex of the questioners was male. We have stated this in the first paragraph of the Results section so that potential readers are aware of participant genders before reading the following Results and Discussion sections. Further, as indicated, we have also discussed the limitation in the Discussion section as follows:

‘Six NT and eight AUT respondents were asked by one of two NT questioners (all men) to share their recent experiences on 12 topics.’

‘Further, this study could not control for sex (gender) differences between groups. This suggests that as well as AUT-specific characteristics, participant-specific characteristics may have influenced our findings.’

Comment 17:

P12, Data Collection, L3 from bottom, P13L3, P13L4, Caption Fig 5, and possibly elsewhere: “experimental paradigm” / “experiment”. Liberal use of “experiment”. An experiment requires dependent and independent variable(s) and allows inference of causal statements. There was no “experiment” here, there were interviews / talks between people.

Response: We thank the reviewer for the pertinent comments. As indicated, we have rephrased it as ‘interview’ throughout the manuscript.

Comment 18:

P13L13, “alacrity”. There is probably a more common synonym for this.

Response: Thank you for the comment. We have reworded it as ‘agile or prompt’.

Comment 19:

P13, last paragraph, L1: “The length of each speech sample was > 10 s. For the sake of replicability of the analysis, the caption of Fig 2 states that there were 4-s excerpts. Were these randomly extracted from the samples of > 10 s? Why, because in this case you could have used samples of at least 4 s, instead of 10 s? Please provide explanation.

Response: We thank the reviewer for their comments. The scalogram was used for a qualitative analysis but not for the quantitative analysis (i.e., statistical analysis such as ANOVA, transfer entropy). For scalogram, the CWT was run on each AM envelope from randomly chosen 4-s ‘excerpts’ derived from the total speech signals. This is because a longer scalogram is difficult to qualitatively unveil each level of phonological rhythm hierarchy (i.e., prosody, syllable, and prosody) and the relationship between the different levels of the phonological hierarchy. Based on the qualitative understanding using scalogram, we then quantitatively analysed two types of speech characteristics using three AM envelopes (prosody: 1‒4 Hz, syllable: 4‒12 Hz, phoneme: 12‒40 Hz). They were described in the Methods section as follows:

‘CWT was run on each AM envelope from randomly chosen 4-s excerpts taken from total speech signals (>10 second) of AUT, NT, AUT-directed, and NT-directed speech. Due to the scalogram being longer, it is difficult to qualitatively unveil each level of the phonological rhythm hierarchy (i.e., prosody, syllable, and prosody) and the relationship between the different levels of the phonological hierarchy.’

‘Next, based on the qualitative understanding using scalogram, we decided the frequency index in AM envelopes (prosody: 1‒4 Hz, syllable: 4‒12 Hz, phoneme: 12‒40 Hz), and quantitatively analysed two types of speech characteristics using three AM envelopes.’

Comment 20:

P14, Data analysis, last paragraph: “The scalograms...chosen 30-s excerpts of speech. In Fig 2, the x-axis denotes time (30 s)...Now it really gets confusing. Both Fig 2 and S3 (which actually states “S1 Appendix”) show scalograms for 4 seconds. Please provide clear and consistent descriptions.

Response: Thank you very much for letting me know of the errors. We corrected them in the Methods section as follows.

CWT was run on each AM envelope from randomly chosen 4-s excerpts taken from total speech signals (>10 second) of AUT, NT, AUT-directed, and NT-directed speech. Due to the scalogram being longer, it is difficult to qualitatively unveil each level of the phonological rhythm hierarchy (i.e., prosody, syllable, and prosody) and the relationship between the different levels of the phonological hierarchy. In Fig 2, the x-axis denotes time (0-4 s), and the y-axis denotes the modulation rate (0.1‒40 Hz).’

Comment 21:

Other than the sampling frequency, there is no description of recording equipment, background noise, etc.

Response: Thank you very much for the comment. As indicated, we have described the recording equipment, background noise, and so on in the Methods section as follows.

‘During speech recording (Marantz, PMD661MKII MP-REC-002, sampling rate = 44.1 kHz, bit rate = 16 bit) through a condenser-type head-worn microphone (SHURE, model BETA54)…’

‘Interviews were conducted inside a sound-proof room to block background noise.’ 

Comment 22:

P15, 2nd paragraph, “...in case the loudness influenced the spectrotemporal modulation feature”. Four lines below this: “in loudness or sound intensity”. Two lines below this: “in pitch and noise”. Pitch and loudness are psychological constructs, frequency and intensity refer to physics, which is likely meant here.

Response: Thank you for the comment. As indicated, we have revised these details in the Methods section as follows:

‘in case the sound intensity influenced the spectrotemporal modulation feature.’

‘AM patterns are implicated in fluctuations in sound intensity’

‘FM patterns reflect fluctuations in spectral frequency’

---

## [Decision Letter · Decision Letter 1]

23 Mar 2023

PONE-D-22-22806R1Non-autistic persons modulate their speech rhythm while talking to autistic individualsPLOS ONE

Dear Dr. Daikoku,

Thank you for submitting your manuscript to PLOS ONE. After careful consideration, we feel that it has merit but does not fully meet PLOS ONE’s publication criteria as it currently stands. Therefore, we invite you to submit a revised version of the manuscript that addresses the points raised during the review process.

We look forward to receiving your revised manuscript.

Kind regards,

Mitsuru Kikuchi, MD, PhD

Academic Editor

PLOS ONE

Journal Requirements:

Reviewers' comments:

Reviewer's Responses to Questions

**Comments to the Author**

1. If the authors have adequately addressed your comments raised in a previous round of review and you feel that this manuscript is now acceptable for publication, you may indicate that here to bypass the “Comments to the Author” section, enter your conflict of interest statement in the “Confidential to Editor” section, and submit your "Accept" recommendation.

Reviewer #1: All comments have been addressed

Reviewer #2: All comments have been addressed

2. Is the manuscript technically sound, and do the data support the conclusions?

Reviewer #1: Yes

Reviewer #2: Yes

3. Has the statistical analysis been performed appropriately and rigorously? 

Reviewer #1: Yes

Reviewer #2: Yes

4. Have the authors made all data underlying the findings in their manuscript fully available?

Reviewer #1: Yes

Reviewer #2: Yes

5. Is the manuscript presented in an intelligible fashion and written in standard English?

Reviewer #1: Yes

Reviewer #2: Yes

6. Review Comments to the Author

Reviewer #1: The authors responded mostly adequately to the reviewer's comments. However, I would recommend a few revisions.

Materials and Methods

Authors should state the official name of MSPA, WAIS-III, Wechsler Memory Scale-revised in Materials and Methods.

Authors should cite which version of the Diagnostic and Statistical Manual of Mental Disorders was used.

Authors should add citations to autism spectrum disorder (ASD) characteristics. Specifically, Autism Spectrum Quotient, Adult Self-Report, and Autism Spectrum Screening Questionnaire were used respectively.

It is helpful for readers to understand the characteristics of the participants if the data such as WAIS3, WMS-R, Autism Spectrum Quotient, Adult Self-Report, and Autism Spectrum Screening Questionnaire, Multi-dimensional Scale for Pervasive Developmental Disorder and Attention Deficit/Hyperactivity Disorder are shown in a table or in the text.

Reviewer #2: The authors have made suitable adjustments to their manuscript, in particular with regard to informing the reader about the nature of the sample characteristics. This gives the article more credibility. The added correlation analysis suggests indeed that NT persons change their way of speaking depending on the speech characteristics of the interlocutor, which corroborates the conclusion. Since the analysis method is original and can be widely applied, the article is at an acceptable level.

Before publication, however, a few (cosmetic) issues need to be fixed:

1. When moving the Materials and Method section to the classic middle position (now from page 6), the authors forgot to fix the Figure order; now Fig. 2 is followed by Fig 6, which should be Fig 3, etc. Captions should be reordered as well.

2. Abstract, Line 26, (all men), also L331 (all men). Since there are just two, "..(both men).." might be better.

3. P6, L147, L148. The acronyms for MSPA, WAIS-III and WMS-R are on P19, L505, 506 and should also be moved to P6.

4. P6, L153, "..about the content of interview.." change to "..the content of the interview.."

5. P16, L428, typo "...d14]." change to "..[14]."

Figure 5, x-axis label "NT Questonnair" change to "NT Questioner"

Figure 5, might consider to simplify the axes values "0" and "2" x10-1

Figure 7 d, "Onset Rime (12-40 Hz)" (??) change to "Phoneme (12-40 Hz)" as in the main text

7. PLOS authors have the option to publish the peer review history of their article (what does this mean?). If published, this will include your full peer review and any attached files.

Reviewer #1: No

Reviewer #2: **Yes: **Gerard B. Remijn

---

## [Author Response · Author response to Decision Letter 1]

20 Apr 2023

RESPONSE TO REVIEWER 1: 

I wish to express our strong appreciation to the Reviewer for the insightful comments on our manuscript. We have studied the Reviewer’s comments very carefully and have made necessary corrections. We feel the comments have helped us significantly improve the manuscript.

Comment 1:

Materials and Methods

Authors should state the official name of MSPA, WAIS-III, Wechsler Memory Scale-revised in Materials and Methods.

Response: Thank you for the comment. We described the official names in the Methods section as follows.

“the Multi-dimensional Scale for Pervasive Developmental Disorder and Attention Deficit/Hyperactivity Disorder (MSPA) was also used to assess the characteristics of developmental disabilities in general. Furthermore, the Wechsler Adult Intelligence Scale-III (WAIS-III) and Wechsler Memory Scale-revised (WMS-R) were used to measure intellectual level and memory function.”

Comment 2:

Authors should cite which version of the Diagnostic and Statistical Manual of Mental Disorders was used.

Response: We thank the reviewer for the pertinent comments. We cited the version as follows.

“The AUT diagnosis was made by a psychiatrist according to the Diagnostic and Statistical Manual of Mental Disorders (DSM-5 [26]).”

Reference

“[26] American Psychiatric Association, D., & American Psychiatric Association. (2013). Diagnostic and statistical manual of mental disorders: DSM-5 (Vol. 5, No. 5). Washington, DC: American psychiatric association.”

Comment 3:

Authors should add citations to autism spectrum disorder (ASD) characteristics. Specifically, Autism Spectrum Quotient, Adult Self-Report, and Autism Spectrum Screening Questionnaire were used respectively.

Response: Thank you for the comment. We cited them as follows.

“the Autism Spectrum Quotient [27], Adult Self-Report [28], and Autism Spectrum Screening Questionnaire [29] were used to measure ASD characteristics.”

Reference

[27] Wakabayashi, A., Tojo, Y., Baron-Cohen, S., & Wheelwright, S. (2004). The Autism-Spectrum Quotient (AQ) Japanese version: evidence from high-functioning clinical group and normal adults. Shinrigaku kenkyu: The Japanese journal of psychology, 75(1), 78-84.

[28] Constantino, J. N., & Gruber, C. P. (2012). Social responsiveness scale: SRS-2 (p. 106). Torrance, CA: Western psychological services.

[29] Ii, T et al.. (2003). 高機能自閉症スペクトラム スクリーニング質問紙 (ASSQ) について. 自閉症と ADHD の子どもたちへの教育支援とアセスメント(English translation: On the high-functioning autism spectrum screening questionnaire (ASSQ). Educational support and assessment for children with autism and ADHD), 39-45. Doi: https://cir.nii.ac.jp/crid/1572261550691517824

Comment 4:

It is helpful for readers to understand the characteristics of the participants if the data such as WAIS3, WMS-R, Autism Spectrum Quotient, Adult Self-Report, and Autism Spectrum Screening Questionnaire, Multi-dimensional Scale for Pervasive Developmental Disorder and Attention Deficit/Hyperactivity Disorder are shown in a table or in the text.

Response: We thank the reviewer for the pertinent comments. 

The publicly available data about characteristics of the participants were summarized into table of Excel spreadsheet and uploaded it to the Open Science Framework for public access (https://osf.io/ayn2w/?view_only=d87230eaeaea428f94464bf633ae4118). Additionally, we have minetined this in the Data Availability section as follows.

“The publicly available data about characteristics of the participants were summarized and have been deposited to an external source (https://osf.io/ayn2w/?view_only=d87230eaeaea428f94464bf633ae4118).”

Data availability

“All the minimal anonymized data set are available in supporting information (S1, S2, S3 and S4 Appendix) and have been deposited to an external source (https://osf.io/ayn2w/?view_only=d87230eaeaea428f94464bf633ae4118).“

 

RESPONSE TO REVIEWER 2:

We are extremely grateful for the Reviewers’ insightful comments on our manuscript. We have studied the Reviewers’ comments carefully and have made the necessary corrections in our paper. We believe that the comments have helped us significantly improve and refine the manuscript.

Comment 1:

1. When moving the Materials and Method section to the classic middle position (now from page 6), the authors forgot to fix the Figure order; now Fig. 2 is followed by Fig 6, which should be Fig 3, etc. Captions should be reordered as well.

Response: We thank the reviewer for the important comments. As indicated, we corrected the order of all figures in whole manuscript.

Comment 2:

Abstract, Line 26, (all men), also L331 (all men). Since there are just two, "..(both men).." might be better.

Response: Thank you for the comment. We revised them as follows.

“two NT questioners (both men)”

Comment 3:

P6, L147, L148. The acronyms for MSPA, WAIS-III and WMS-R are on P19, L505, 506 and should also be moved to P6.

Response: We thank the reviewer for comments. As indicated, we stated these full spells in the Methods section and the acronyms were moved to the Discussion section as follows.

Methods section

“the Multi-dimensional Scale for Pervasive Developmental Disorder and Attention Deficit/Hyperactivity Disorder (MSPA) was also used to assess the characteristics of developmental disabilities in general. Furthermore, the Wechsler Adult Intelligence Scale-III (WAIS-III) and Wechsler Memory Scale-revised (WMS-R) were used to measure intellectual level and memory function.”

Discussion section

“This study also checked the MSPA to control the characteristics of developmental disabilities in general. Furthermore, the WAIS-III and WMS-R were used to control intellectual level and memory function.”

Comment 4:

P6, L153, "..about the content of interview.." change to "..the content of the interview.."

Response: As indicated, we revised it as follows.

“All participants were informed the content of the interview”

Comment 5:

P16, L428, typo "...d14]." change to "..[14]."

Response: We corrected it as follows.

“enhancing speech learnability [14].”

Comment 6:

Figure 5, x-axis label "NT Questonnair" change to "NT Questioner"

Figure 5, might consider to simplify the axes values "0" and "2" x10-1

Response: Thank you for letting me know the typo. We also simplified the axes values "0" and "2" x10-1 to the axes values 0 and 0.2. We revised them as follows.

Comment 7:

Figure 7 d, "Onset Rime (12-40 Hz)" (??) change to "Phoneme (12-40 Hz)" as in the main text

Response: Thank you for letting me know the typo. We revised it as follows.

---

## [Decision Letter · Decision Letter 2]

27 Apr 2023

Non-autistic persons modulate their speech rhythm while talking to autistic individuals

PONE-D-22-22806R2

Dear Dr. Daikoku,

We’re pleased to inform you that your manuscript has been judged scientifically suitable for publication and will be formally accepted for publication once it meets all outstanding technical requirements.

Kind regards,

Mitsuru Kikuchi, MD, PhD

Academic Editor

PLOS ONE

Additional Editor Comments (optional):

Reviewers' comments:

Reviewer's Responses to Questions

**Comments to the Author**

1. If the authors have adequately addressed your comments raised in a previous round of review and you feel that this manuscript is now acceptable for publication, you may indicate that here to bypass the “Comments to the Author” section, enter your conflict of interest statement in the “Confidential to Editor” section, and submit your "Accept" recommendation.

Reviewer #1: (No Response)

Reviewer #2: All comments have been addressed

2. Is the manuscript technically sound, and do the data support the conclusions?

Reviewer #1: (No Response)

Reviewer #2: Yes

3. Has the statistical analysis been performed appropriately and rigorously? 

Reviewer #1: (No Response)

Reviewer #2: Yes

4. Have the authors made all data underlying the findings in their manuscript fully available?

Reviewer #1: (No Response)

Reviewer #2: Yes

5. Is the manuscript presented in an intelligible fashion and written in standard English?

Reviewer #1: Yes

Reviewer #2: Yes

6. Review Comments to the Author

Reviewer #1: (No Response)

Reviewer #2: The manuscript can be accepted, in my opinion.

One small issue: in the 2nd review round I suggested to add "the" before "interview", but I forgot to add "about" to the corrected part. So please change:

P7, L160 "All participants were informed the content of the interview...." change to "All participants were informed about the content of the interview...."

Sorry about this.

7. PLOS authors have the option to publish the peer review history of their article (what does this mean?). If published, this will include your full peer review and any attached files.

Reviewer #1: No

Reviewer #2: **Yes: **Gerard B. Remijn

---

## [Editor Report · Acceptance letter]

2 May 2023

PONE-D-22-22806R2 

Non-autistic persons modulate their speech rhythm while talking to autistic individuals 

Dear Dr. Daikoku:

I'm pleased to inform you that your manuscript has been deemed suitable for publication in PLOS ONE. Congratulations! Your manuscript is now with our production department. 

Kind regards, 

on behalf of

Dr. Mitsuru Kikuchi 

Academic Editor

PLOS ONE